# PRISM: A HIERARCHICAL MULTISCALE APPROACH FOR TIME SERIES FORECASTING

## ABSTRACT

Forecasting is critical in areas such as finance, biology, and healthcare. Despite the progress in the field, making accurate forecasts remains challenging because real-world time series contain both global trends, local fine-grained structure, and features on multiple scales in between. Here, we present a new forecasting method, PRISM (Partitioned Representation for Iterative Sequence Modeling), that addresses this challenge through a learnable tree-based partitioning of the signal. At the root of the tree, a global representation captures coarse trends in the signal, while recursive splits reveal increasingly localized views of the signal. At each level of the tree, data are projected onto a time-frequency basis (e.g., wavelets or exponential moving averages) to extract scale-specific features, which are then aggregated across the hierarchy. This design allows the model to jointly capture global structure and local dynamics, enabling both reconstruction and forecasting. Experiments across benchmark datasets show that our method outperforms state-of-the-art methods for forecasting and also requires less runtime and memory. Overall, these results demonstrate that our hierarchical approach provides a lightweight and flexible framework for forecasting multivariate time series. The code [will be] available in [anonymized].

## 1 INTRODUCTION

The ability to anticipate the future is one of the most valuable tools in science and society. From natural systems to human behavior, prediction allows us to plan, adapt, and intervene before events unfold (1; 3; 33). Time series forecasting provides a formal way to make such predictions, but it remains challenging because signals evolve across multiple scales (1). Global trends interact with seasonal rhythms and local fluctuations, creating a hierarchical structure in which information at one scale sets context for the next. We argue that effective forecast models must capture this hierarchy explicitly, aligning long-term dynamics with fine-scale variability in a coherent representation.

Recent work has moved toward this goal, but current approaches fall short of learning unified hierarchical time-frequency representations. Some models incorporate frequency cues to capture periodicity and compact spectral structure (25; 32), others emphasize temporal hierarchy through coarse-to-fine decompositions (2; 26), and still others operate mainly in the frequency domain (27; 28) without modeling the temporal structure. While these directions highlight the value of hierarchical design, they construct it in only one domain at a time or flatten frequency into shallow features without linking it to temporal context. What remains missing is a coherent and reconstructable hierarchy that organizes both time and frequency, and captures long-term structure, short-term variability, and the interactions between them in a unified representation.

In this work, we introduce PRISM (*Partitioned Representations for Iterative Sequence Modeling*), a new model that fills this gap by jointly learning a hierarchical decomposition in both time and frequency. PRISM builds a binary tree over time, recursively partitioning the input into overlapping segments that preserve local context. At each node, a time-frequency decomposition (via Haar wavelets (15) or alternative transforms like exponential moving average filters) produces multiple bands, which are then adaptively weighted by a lightweight router that reweighs bands. The outputs of child nodes are stitched together through linear mixing, yielding stable reconstructions and multiscale representations that are both interpretable and effective for forecasting. This design provides a

unified tree over time and frequency, with an explicit reconstruction pathway that grounds forecasts in coherent signal structure.

Across benchmark datasets, `PRISM` achieves accuracy competitive with strong state-of-the-art baselines and surpasses them in many forecasting settings. The model also provides interpretability through its frequency-weighting mechanism, revealing which bands contribute most to predictions, and robustness through its reconstructible design, which stabilizes learning and scales naturally to longer contexts. Together, these results demonstrate that hierarchical multiscale representations, when organized into a unified time–frequency tree, provide a principled pathway toward more accurate and efficient forecasting.

- We propose `PRISM`, a forecasting architecture that organizes time series data through a binary temporal hierarchy combined with multiresolution frequency encoding. This structure captures dependencies across temporal and spectral scales, linking long-range trends with short-term fluctuations.
- `PRISM` learns importance scores over all time–frequency nodes, enabling the model to focus on the most predictive components. A lightweight auxiliary reconstruction loss further stabilizes learning and promotes interpretable importance patterns.
- Through extensive evaluation across diverse datasets and horizons, `PRISM` demonstrates strong generalization and efficiency. The model consistently outperforms competitive baselines while maintaining a compact design and transparent interpretability.

## 2  RELATED WORK

We group prior work by how they use time and frequency information. We first review models that process both domains at once. We then cover models that build a hierarchy only in time. Next we discuss methods that work mainly in the frequency domain. We conclude this section by positioning our approach among these prior works.

**Joint time and frequency modeling methods.**    Several methods combine information from the time and frequency domains at the same time. TimesNet (25) first discovers dominant periods with an FFT search. It reshapes a 1D series into 2D period–phase tensors and applies 2D CNN blocks to capture patterns inside each period and across periods. This gives a compact time stack guided by frequency peaks. FEDformer (32) adds a seasonal–trend decomposition to a Transformer. It replaces standard attention with frequency-enhanced blocks that operate on a sparse set of Fourier or wavelet modes. This reduces cost and focuses the model on informative bands. ETSformer (24) builds on exponential smoothing and adds frequency attention. It organizes the network into level, growth, and seasonality modules, so each module has a clear role. TFDNet (11) forms a time–frequency matrix with the short-time Fourier transform (STFT), then uses multi-scale encoders and separate time and frequency blocks for trend and seasonal parts. This design lets the model align slow and fast dynamics across domains. These models mix domains and gain efficiency from compact spectral views. However, they do not build a learnable hierarchy in both time and frequency, and they do not define a reconstructable pathway that stitches local pieces back together. Another work is CoST (23), which uses time-domain and frequency-domain contrastive losses to learn disentangled trend and seasonal representations for downstream forecasting

**Time domain hierarchical decomposition methods.**    A second line of work builds a multiscale structure only along time. Early statistical models such as ARIMA (1) and SARIMA (1) decompose a series into trend, seasonality, and noise with fixed forms and linear dynamics.

Later deep learning methods keep the idea of decomposition but learn it from data. For instance: N-HiTS (2) uses hierarchical interpolation and multi-rate sampling. It assembles forecasts at several resolutions, so coarse blocks capture slow movements and fine blocks refine details. DLinear (29) is another notable work which uses seasonal-trend decomposition integrated with linear layers, maintaining a simple structure with impressive performance. TimeMixer (21) is an MLP forecaster that performs decomposable multi-scale mixing on the past. It also ensembles several simple predictors for the future to improve stability. SCINet (8) applies a recursive split with downsampling, convolutions, and cross-branch interaction. The split-and-interact pattern captures multi-resolution patterns while keeping computation low. N-BEATS (14) is a basis-expansion stack with backcast and forecast heads. The interpretable variant uses polynomial and harmonic bases to separate trend and

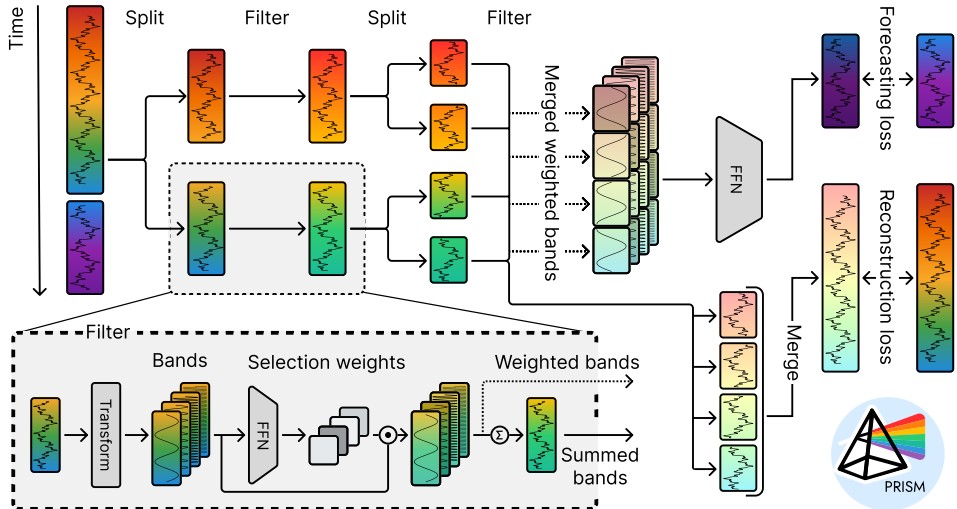

Figure 1: **The PRISM model overview.** The timeseries is partitioned into chunks recursively to produce smaller temporal segments. At each level of this splitting procedure, we apply learnable filter banks that use time-frequency representations and learnable weights to extract features from the segment. The weights at each level of the feature hierarchy are distinct and not shared, allowing the model to extract different sets of coefficients that are meaningful for forecasting and reconstruction. The learning is driven by two objectives, a forecasting loss (top, right) on predictions of the future and a reconstruction loss on the past (bottom, right).

seasonality. Residual stacking gives a deep hierarchy over time. Autoformer (26) places a progressive trend–seasonal decomposition inside the network. An auto-correlation module models long-range dependencies in a cost-effective way. MICN (20) uses multiple convolutional branches with different kernels. A local branch with downsampling convolution captures short-range cues, while a global branch with isometric convolution covers long context. Pyraformer (9) introduces pyramidal attention. An inter-scale tree summarizes features at several resolutions with low complexity, which helps on long sequences. These designs learn a clear hierarchy over time. They do not pair it with a matching frequency hierarchy, and they do not route information across bands at each time scale.

**Multiscale frequency modeling or selection methods.** A third line of work focuses on the frequency side. FreTS (27) transforms a series to the frequency domain and applies redesigned MLPs to the real and imaginary parts of these coefficients to give a compact global spectral representation. FiLM (31) uses Legendre projections to form a compact long memory for the history and Fourier projections to reduce noise before using a light expert mixer to combine predictions from multi-scale views of the history. D-PAD (28) splits a series into frequency ranges with a multi-component decomposition. It then applies a decomposition–reconstruction–decomposition pipeline to disentangle information at the same frequency and fuses the components for forecasting. These methods emphasize spectral selection and disentangling. They do not build a time hierarchy and they do not define a joint tree over both domains.

**Positioning of our method.** Our method, PRISM, learns a hierarchy in both domains at the same time. It builds a binary tree over time with overlap to keep local context and smooth boundaries. At each node it applies a same-length, reconstructable band partition. A lightweight router selects bands per node and forwards a band-weighted sum to the left and right children. Child outputs are stitched with a linear cross-fade, so reconstruction is simple and stable. This yields a coherent time–frequency hierarchy that supports both reconstruction and forecasting, and it fills the gap left by prior work that is hierarchical in only one domain or that mixes domains without a shared hierarchy.

## 3 METHODS

Here we describe PRISM, our model for the forecasting of the time series. We train our model on a combination of two losses, reconstruction of the past and prediction of the future. We provide an overview of our method in Figure 1.

## 3.1 PROBLEM SETUP

We define the task of time series forecasting as follows (12): Let $\mathcal{X} = \{x_t\}_{t=1}^T$ represent a univariate or multivariate time series in input space $\mathbb{R}^D$, where $t \in \{1, 2, \ldots, T\}$ represents time. We consider a context window $\mathcal{X}_{\text{context}} = \{x_t\}_{t=1}^{T_{\text{context}}}$ of length $T_{\text{context}}$ and a prediction window $\mathcal{X}_{\text{forecast}} = \{x_t\}_{t=T-T_{\text{forecast}}}^T$ of length $T_{\text{forecast}}$ such that $T_{\text{context}} + T_{\text{forecast}} = T$. The objective is then to build a model that given the context window $\mathcal{X}_{\text{context}}$, predicts the time series in the prediction window $\mathcal{X}_{\text{forecast}}$. To prepare the data for our model, we sample segments $x \in \mathbb{R}^{B \times T \times C}$ from raw time series provided in datasets. Here $B$ denotes batch size, $C$ denotes the number of the recorded channels in a multivariate time series, and $T$ denotes the length (duration) of the time series.

## 3.2 THE PRISM MODEL

In our PRISM model, the data is cycled through (i) the time decomposition step; (ii) the frequency decomposition step, (iii) the importance weighting of frequency bands, and (iv) the reconstruction step, thus generating hierarchical representations of input time series both in time and frequency domains. The temporal hierarchy is obtained through iterated bisection of the time series; thus, this hierarchy is not formed in a single step but requires the entire iteration. The frequency hierarchy, in contrast, is obtained separately for every level of the time hierarchy. It provides signal decomposition, progressively refining the input for the next time-decomposition steps. We provide the details below.

In the **time decomposition step** $i_{\text{temp}}$, we split the time series of the length (duration) $T(i_{\text{temp}})$ (dimensions: $T(i_{\text{temp}}) \times C$; here and below we omit the batch dimension $B$) along the time axis into two equally sized segments with the overlap $o$ (resulting durations: $T(i_{\text{temp}} + 1) = (T(i_{\text{temp}}) + o)/2$). On the first iteration of this step, the duration $T(i_{\text{temp}} = 0)$ equals to the length of the model's context window $T_{\text{context}}$; on the second iteration it equals to $(T_{\text{context}} + o)/2$ and so on. This sequential process produces a hierarchy of overlapping segments of the time series, with $M = 2^{i_{\text{temp}}}$ segments on the iteration $i_{\text{temp}}$, in the form of the binary tree.

Each time decomposition step is followed by the **frequency decomposition step** where we use the Haar discrete wavelet transformation (15) (DWT; $K=6$) to split each segment of the time series (whose size $T(i_{\text{temp}}) \times C$ varies by the iteration $i_{\text{temp}}$) into the hierarchy of $K$ signals corresponding to different frequency bands of the original signal (overall dimension: $T(i_{\text{temp}}) \times C \times K$). The hierarchical frequency filters here are easily interchangeable. In ablations, we replace Haar DWT with the fast Fourier transformation (16; 19) (FFT; $K=4$, rectangular masks), binomial pyramids (5) ($K=4$, $k_0=3$, $k_{\text{grow}}=2$), differences of Gaussians (7) (DoG; $K=6$, $\sigma_0=1.0$, ratio=1.6), exponential moving averages (4) (EMA; $K=4$, $\tau_0=8.0$, grow=3.0), and MCD block from D-PAD (28).

To calibrate the impact of frequency bands in time segments of the signal, we follow each frequency decomposition step with the **importance weighting** step. To this end, we compute the summary statistics for each frequency band of every time segment of the signal $(\mu, \sigma, a_{\text{max}}, \|\Delta\|_1, \|\Delta^2\|_1, a_{\text{max}}/(\sigma+\varepsilon))$ and pass it as an input to a two-layer MLP. Here $\mu$ is the mean value, $\sigma$ is the standard deviation, and $a_{\text{max}}$ is the maximum amplitude of the signal; $\|\cdot\|_1$ is the $l_1$-norm; $\Delta$ and $\Delta^2$ are the first and the second derivatives of the signal. Using the summary statistics of the signal as opposed to the signal itself as input reduces the computational complexity of the model and allows for the shared use of the same MLP with signals of different length (e.g. weight sharing across the scales of time hierarchy). The MLP produces a single importance score $s_{c,k}$ per input (here the channel $c \in C$ and the frequency band $k \in K$). The importance scores $s_{c,k}$ are then converted into the importance weights $w_{c,k}$ for the frequency bands through the softmax operation with temperature $\tau$.

These importance weight are used in the subsequent **reconstruction step** to combine the $K$ signals (of size $T(i_{\text{temp}} + 1) \times C$) from all the frequency bands into the joint signal (of size $T(i_{\text{temp}} + 1) \times C$). This way, the joint signal retains the information that is most important for the downstream forecasting task, while irrelevant information is suppressed. All the steps described above are then iterated, resulting in the processing of progressively smaller segment of time series, enabling multiscale analysis of its data. On the last iteration, prior to combining individual bands into the joint signal, they are concatenated along the time dimension with linear cross-fade over the overlap window $o$ to form the sequence of the original size $T_{\text{context}} \times C$.

The concatenated multi-band signal (of size $B \times T_{\text{context}} \times C \times K$) is then fed as input to $M \times K$ two-layer MLPs (for $M$ temporal segments times $K$ frequency segments) that outputs the forecast

for the time series over the forecast window ($\chi_{\text{forecast}}^{\text{model}}$ of size $B \times T_{\text{forecast}} \times C$). The other version of the multi-band signal, summed over the band dimension ($\chi_{\text{context}}^{\text{model}}$ of size $B \times T_{\text{context}} \times C$), is meant to reconstruct the time series signal over the context window. Both outputs are used in the loss function as described below. We train the model end-to-end with an MSE loss function that couples the forecasting loss with the auxiliary reconstruction loss:

$$\mathcal{L} = \text{MSE}(\chi_{\text{forecast}}^{\text{model}}, \chi_{\text{forecast}}) + \text{MSE}(\chi_{\text{context}}^{\text{model}}, \chi_{\text{context}}). \tag{1}$$

We train our model in batches of $B = 512$ using the Adam optimizer with learning rate 1e-4. Early stopping monitors the validation MSE with the patience of 15 steps and the improvement margin of $\delta = 2 \times 10^{-4}$. We evaluate the model on the held-out test split to report final MSE and MAE.

## 4 RESULTS

### 4.1 EXPERIMENT SETUP

**Datasets.** To comprehensively examine the performance of our model under diverse settings, we have tested it on a standard set of time-series datasets (25). The *ETT* datasets (30) (*ETTh1, ETTh2, ETTm1, ETTm2*) record electricity transformer temperature data collected either hourly or every 15 minutes from 2016 to 2018, each including seven variables such as oil temperature and load. The *Electricity* dataset (6) consists of hourly electricity consumption records for 321 customers between 2012 and 2014, while the *Traffic* dataset contains hourly road occupancy rates measured by 862 sensors across the San Francisco Bay Area. The *Exchange* dataset (6) consists of daily exchange rates of eight currencies against the US dollar, collected from 1990 to 2016. Additionally, the *Weather* dataset (29) includes 21 meteorological indicators collected every 10 minutes from a weather station in 2020. For the consistency with prior literature, we followed the established protocols by splitting the *ETT* datasets into training, validation, and test sets using a 6:2:2 ratio, and applied a 7:1:2 split for the remaining datasets (22). This panel of diverse benchmarks has enabled the assessment of our model's forecasting capabilities across varying temporal resolutions and signal characteristics (25).

**Evaluation.** To contextualize our results, we compared the accuracy of the forecasts of our model to the accuracies of prominent other models available in literature. To evaluate the models under varied settings, we trained them to make predictions over varied time horizons including 96, 192, 336, and 720 timesteps into the future. We optimized the hyperparameters of our model on the ETT datasets (Supplementary Tables 9, 10, 11). As the models in prior work were often trained and evaluated only on subsets of these benchmarks and time horizons, we have reevaluated these models to provide the comprehensive results in all the considered settings.

### 4.2 PERFORMANCE ON FORECASTING BENCHMARKS

**`PRISM` sets overall SOTA performance on a panel of standard datasets.** In this section, we have compared the performance of our model to that of several highest-performing methods including D-PAD (28) (current state-of-the-art method for timeseries forecasting), DLinear (29), TimeMixer (21), N-HiTS (2), iTransformer (10), PatchTST (13), and XPatch (18) (Table 1). We used the total of 8 datasets, considering 4 forecast horizons for each of them ($4 \times 8 = 32$ evaluations total), as outlined above. We found that, out of 32 considered setting, our model, `PRISM`, has shown the best MSE in 17 settings and the best MAE in 18 settings, consistently outperforming all the models within the comparison. Our model was followed by D-PAD, a model that has shown the best MSE in 9 settings and the best MSE in 10 settings. We provide additional resuls for a different context length in Supplementary Table 12. Notably, D-PAD is a model that utilizes a hierarchical temporal encoder but does not use a hierarchical frequency encoder. We provide additional results for where we compare `PRISM` with the closest competitors (D-PAD, DLinear) on the GIFT (17) benchmark (Supplementary Table 5). On GIFT, `PRISM` has shown the best MSE in 61 datasets; D-PAD was the best on 21 datasets, and DLinear was the best on 9 datasets. For MAE, the corresponding numbers were: 52, 25, and 14 datasets, again favoring `PRISM`.

**`PRISM` shows the best overall performance on irregular, aperiodic, incomplete, nonstationary, and drifting data.** To test how our model handles different types of artifacts in time series data, we have evaluated its performance on irregular, aperiodic, incomplete, nonstationary, anomalous, and

Table 1: **Forecasting performance across varying future horizons.** The MSE and MAE are reported for 8 datasets over 4 seeds. We compare the performance of our method PRISM with D-PAD (28), DLinear (29), TimeMixer, N-HiTS (2), and transformer-based baselines (iTransformer (10), PatchTST (13), and XPatch (18)). The Context window is 336 for all datasets and the horizons are 96, 192, 336, 720 samples.

| Dataset | Ctx | H | PRISM | | D-PAD | | DLinear | | TimeMixer | | N-HiTS | | iTransformer | | PatchTST | | XPatch | |
|---|---|---|---|---|---|---|---|---|---|---|---|---|---|---|---|---|---|---|
| | | | MSE | MAE | MSE | MAE | MSE | MAE | MSE | MAE | MSE | MAE | MSE | MAE | MSE | MAE | MSE | MAE |
| ETTh1 | 336 | 96 | 0.355 | 0.374 | 0.357 | 0.376 | 0.375 | 0.399 | 0.388 | 0.410 | 0.475 | 0.498 | 0.386 | 0.405 | 0.414 | 0.419 | 0.376 | 0.386 |
| | 336 | 192 | 0.390 | 0.405 | 0.394 | 0.402 | 0.405 | 0.416 | 0.420 | 0.432 | 0.492 | 0.519 | 0.441 | 0.436 | 0.413 | 0.429 | 0.417 | 0.407 |
| | 336 | 336 | 0.386 | 0.412 | 0.374 | 0.412 | 0.479 | 0.443 | 0.487 | 0.466 | 0.550 | 0.564 | 0.487 | 0.458 | 0.422 | 0.440 | 0.449 | 0.425 |
| | 336 | 720 | 0.421 | 0.445 | 0.419 | 0.442 | 0.472 | 0.490 | 0.503 | 0.481 | 0.598 | 0.641 | 0.503 | 0.491 | 0.447 | 0.468 | 0.470 | 0.456 |
| ETTh2 | 336 | 96 | 0.267 | 0.322 | 0.272 | 0.327 | 0.289 | 0.353 | 0.283 | 0.343 | 0.328 | 0.364 | 0.297 | 0.349 | 0.274 | 0.337 | 0.233 | 0.300 |
| | 336 | 192 | 0.311 | 0.359 | 0.331 | 0.368 | 0.383 | 0.418 | 0.350 | 0.390 | 0.372 | 0.408 | 0.380 | 0.400 | 0.341 | 0.382 | 0.291 | 0.338 |
| | 336 | 336 | 0.318 | 0.364 | 0.321 | 0.370 | 0.448 | 0.465 | 0.389 | 0.423 | 0.397 | 0.421 | 0.428 | 0.432 | 0.329 | 0.384 | 0.344 | 0.377 |
| | 336 | 720 | 0.390 | 0.421 | 0.399 | 0.426 | 0.605 | 0.551 | 0.443 | 0.458 | 0.461 | 0.497 | 0.427 | 0.445 | 0.379 | 0.422 | 0.407 | 0.427 |
| ETTm1 | 336 | 96 | 0.288 | 0.321 | 0.285 | 0.328 | 0.299 | 0.343 | 0.299 | 0.352 | 0.370 | 0.468 | 0.334 | 0.368 | 0.293 | 0.346 | 0.311 | 0.346 |
| | 336 | 192 | 0.325 | 0.359 | 0.323 | 0.349 | 0.335 | 0.365 | 0.336 | 0.375 | 0.436 | 0.488 | 0.377 | 0.391 | 0.333 | 0.370 | 0.348 | 0.368 |
| | 336 | 336 | 0.358 | 0.379 | 0.351 | 0.372 | 0.369 | 0.386 | 0.379 | 0.406 | 0.483 | 0.510 | 0.426 | 0.420 | 0.369 | 0.392 | 0.388 | 0.391 |
| | 336 | 720 | 0.410 | 0.419 | 0.412 | 0.405 | 0.425 | 0.421 | 0.432 | 0.436 | 0.489 | 0.537 | 0.491 | 0.459 | 0.416 | 0.420 | 0.461 | 0.430 |
| ETTm2 | 336 | 96 | 0.158 | 0.234 | 0.162 | 0.247 | 0.167 | 0.263 | 0.178 | 0.262 | 0.184 | 0.262 | 0.180 | 0.264 | 0.166 | 0.256 | 0.164 | 0.248 |
| | 336 | 192 | 0.214 | 0.267 | 0.218 | 0.283 | 0.224 | 0.303 | 0.239 | 0.304 | 0.260 | 0.293 | 0.250 | 0.309 | 0.223 | 0.296 | 0.230 | 0.291 |
| | 336 | 336 | 0.264 | 0.327 | 0.267 | 0.321 | 0.284 | 0.342 | 0.281 | 0.334 | 0.313 | 0.359 | 0.311 | 0.348 | 0.274 | 0.329 | 0.292 | 0.331 |
| | 336 | 720 | 0.360 | 0.383 | 0.353 | 0.372 | 0.397 | 0.421 | 0.363 | 0.388 | 0.411 | 0.421 | 0.412 | 0.407 | 0.362 | 0.385 | 0.381 | 0.383 |
| Traffic | 336 | 96 | 0.362 | 0.237 | 0.359 | 0.236 | 0.410 | 0.282 | 0.380 | 0.275 | 0.407 | 0.290 | 0.395 | 0.268 | 0.360 | 0.249 | 0.481 | 0.280 |
| | 336 | 192 | 0.376 | 0.244 | 0.377 | 0.245 | 0.423 | 0.287 | 0.401 | 0.282 | 0.423 | 0.302 | 0.417 | 0.276 | 0.379 | 0.256 | 0.484 | 0.275 |
| | 336 | 336 | 0.401 | 0.257 | 0.391 | 0.253 | 0.436 | 0.296 | 0.413 | 0.290 | 0.446 | 0.321 | 0.433 | 0.283 | 0.392 | 0.264 | 0.500 | 0.279 |
| | 336 | 720 | 0.433 | 0.271 | 0.413 | 0.282 | 0.446 | 0.305 | 0.455 | 0.305 | 0.528 | 0.369 | 0.467 | 0.302 | 0.432 | 0.286 | 0.534 | 0.293 |
| Electricity | 336 | 96 | 0.122 | 0.219 | 0.128 | 0.220 | 0.140 | 0.237 | 0.130 | 0.224 | 0.151 | 0.254 | 0.148 | 0.240 | 0.129 | 0.222 | 0.159 | 0.244 |
| | 336 | 192 | 0.145 | 0.235 | 0.148 | 0.236 | 0.153 | 0.247 | 0.152 | 0.246 | 0.170 | 0.273 | 0.149 | 0.231 | 0.147 | 0.240 | 0.160 | 0.248 |
| | 336 | 336 | 0.159 | 0.243 | 0.163 | 0.253 | 0.160 | 0.259 | 0.169 | 0.263 | 0.200 | 0.291 | 0.178 | 0.253 | 0.163 | 0.259 | 0.182 | 0.267 |
| | 336 | 720 | 0.194 | 0.281 | 0.201 | 0.286 | 0.203 | 0.301 | 0.204 | 0.293 | 0.244 | 0.356 | 0.225 | 0.317 | 0.197 | 0.290 | 0.216 | 0.298 |
| Exchange | 336 | 96 | 0.081 | 0.198 | 0.098 | 0.219 | 0.082 | 0.202 | 0.089 | 0.211 | 0.092 | 0.208 | 0.086 | 0.206 | 0.095 | 0.217 | 0.082 | 0.199 |
| | 336 | 192 | 0.176 | 0.297 | 0.204 | 0.321 | 0.157 | 0.293 | 0.194 | 0.315 | 0.208 | 0.300 | 0.177 | 0.299 | 0.201 | 0.322 | 0.177 | 0.298 |
| | 336 | 336 | 0.314 | 0.396 | 0.429 | 0.464 | 0.305 | 0.414 | 0.351 | 0.429 | 0.371 | 0.509 | 0.331 | 0.417 | 0.372 | 0.447 | 0.349 | 0.425 |
| | 336 | 720 | 0.712 | 0.606 | 0.721 | 0.607 | 0.643 | 0.601 | 1.019 | 0.744 | 0.888 | 1.447 | 0.847 | 0.691 | 0.873 | 0.699 | 0.891 | 0.711 |
| Weather | 336 | 96 | 0.140 | 0.177 | 0.143 | 0.181 | 0.176 | 0.237 | 0.152 | 0.206 | 0.160 | 0.197 | 0.174 | 0.214 | 0.149 | 0.198 | 0.146 | 0.184 |
| | 336 | 192 | 0.187 | 0.227 | 0.189 | 0.229 | 0.220 | 0.282 | 0.197 | 0.255 | 0.207 | 0.265 | 0.221 | 0.254 | 0.194 | 0.241 | 0.190 | 0.228 |
| | 336 | 336 | 0.234 | 0.266 | 0.239 | 0.268 | 0.265 | 0.319 | 0.249 | 0.283 | 0.273 | 0.301 | 0.278 | 0.296 | 0.245 | 0.282 | 0.236 | 0.273 |
| | 336 | 720 | 0.302 | 0.311 | 0.304 | 0.313 | 0.323 | 0.362 | 0.322 | 0.338 | 0.363 | 0.352 | 0.358 | 0.347 | 0.314 | 0.334 | 0.309 | 0.321 |
| *Best count (MSE)* | | | 17 | — | 9 | — | 3 | — | — | — | — | — | — | — | 1 | — | 2 | — |
| *Best count (MAE)* | | | — | 18 | — | 10 | — | 3 | — | — | — | — | — | — | — | 0 | — | 2 |

drifting data. To this end, we used the GIFT dataset: By having nearly 100 datasets, it naturally has multiple datasets featuring each of the artifacts of interest. We compared the performance of our model with two closest competitors (D-PAD, DLinear; Table 2), as identified in the section above.

To represent **irregular** datasets, featuring irregular timestamps or asynchronous sensors, we used *bitbrains* (cloud workload traces, asynchronous VM sensors), *bizitobs* (business telemetry, event-driven metrics), *solar/10T* (where sensor outages have causes irregular intervals), and *loop_seattle* (traffic loops with packet loss / clock drift). We found that our model PRISM has shown the best performance on 16 datasets (MSE) and on 14 datasets (MAE), followed by D-PAD (6 and 7 datasets respectively).

For **aperiodic** datasets, we chose the ones that lack fixed seasonal structure or represent stochastic / chaotic signals including *exchange-rate*, *bitbrains*, *bizitobs*, *solar/H/* (short-term irradiance fluctuations), *hospital*, *covid_deaths*, and *us_births* (event-driven, non-stationary), as well as the *kdd_cup_2018* (network KPI with bursty events). All these datasets were either dominated by random events or regime changes where standard daily/weekly seasonality broke down. In these tests, PRISM has shown the best results on 20 datasets (MSE) and 18 datasets (MAE), followed by DLinear (6 and 7 datasets respectively).

To analyze the impact of the **incomplete** observations on model's performance, we chose the datasets featuring sensor dropouts, partial series, or unavailable covariates. To this end, we used *solar*, *electricity*, *weather/jena*, *loop_seattle*; *traffic*, *sz_taxi* (as urban sensors were often missing), and *bizitobs* (featuring irregular ingestion logs). Some of these datasets have specified the missing ratios of 5–30%. For incomplete data, PRISM came first featuring the best performance on 27 datasets (MSE) and 24 datasets (MAE), followed by D-PAD (10 and 13 datasets).

To study how **anomalies** affect the forecasting performance, we considered the datasets with transient spikes, hardware faults, or external shocks. These included *traffic/loop_seattle*, *sz_taxi*, and *m4* (for holiday spikes); *solar/H/short*, andsolar/W/short (for cloud passage spikes); *bitbrains_rnd*, and

*bizitobs_service* (for burst loads); finally, *exchange-rate*, and *crypto* (for financial jumps). These datasets have featured large deviations in residuals, high kurtosis, and occasional sign flips. This test was dominated by the `PRISM` with the best performance on 13 datasets (MSE) and 11 datasets (MAE), followed bY D-PAD (8 and 8 datasets).

To investigate the **nonstationarity** in the data, we used datasets whose input distributions changed across time including day/night oscillations, seasonal impact, or market regimes. These datasets included *ett1*, *ett2*, *electricity*, *jena_weather*; *traffic/loop_seattle* and *sz_taxi* (for rush-hour pattern drift); *solar* (featuring daily to seasonal irradiance shifts); lastly, *us_births* and *hospital* (for the demographic or policy drift). Such datasets are viewed as challenging, as the training on early segments of data may lead to the decreased performance on later ones unless adaptive normalization or re-weighting is used. In this task, `PRISM` has shown the best results on 33 datasets (MSE) and 29 datasets (MAE), followed by D-PAD (11 and 15 datasets).

Finally, to look into the effects of the long-term **drift** in the data, we selected the datasets featuring a slow trend or a persistent distributional movement: *weather/jena_weather*, *temperature_rain*, *saugeen* (for climate drift); *electricity/H/long* and *energy/ETT-long* (for economic growth trend); *us_births* and *covid_deaths* (for population or pandemic phase change); also *traffic/METR-LA* and *sz_taxi* (for infrastructure evolution). These datasets have featured a monotonic mean/variance shift and/or gradual phase migration in frequency spectra. On this last task, `PRISM` came first with the best results on 17 datasets (MSE) and 15 datasets (MAE), followed by D-PAD (4 and 6 datasets respectively). Thus, our model `PRISM` has shown the best results on 5 out of 6 types of data.

Table 2: **Forecasting performance across different types of data.** Here we present the numbers of the datasets on which each respective model has shown the best performance.

| Model | Overall | | Irregular | | Aperiodic | | Incomplete | | Anomaly | | Nonstationary | | Drift | |
|---|---|---|---|---|---|---|---|---|---|---|---|---|---|---|
| | MSE | MAE | MSE | MAE | MSE | MAE | MSE | MAE | MSE | MAE | MSE | MAE | MSE | MAE |
| PRISM | **61** | **52** | **16** | **14** | **20** | **18** | **27** | **24** | **13** | **11** | **33** | **29** | **17** | **15** |
| D-PAD | 21 | 25 | 6 | 7 | 4 | 5 | 10 | 13 | 8 | 8 | 11 | 15 | 4 | 6 |
| DLinear | 9 | 14 | 4 | 5 | 6 | 7 | 5 | 5 | 3 | 5 | 6 | 6 | 2 | 2 |

Overall, our results speak to the utility of jointly using hierarchical time and frequency decompositions in the forecasting of time series. We further investigate this proposal in ablation sections below.

## 4.3 ABLATIONS

To verify that each design choice is necessary and that the overall configuration is optimized for time-series forecasting, we conduct targeted ablations that remove or alter each of the model's component in turns. Our `PRISM` model consists of a hierarchical tree encoder for the time domain, (hierarchical) feature decomposition for the frequency domain, learnable importance scoring for frequency features, and the joint loss that encompasses forecasting and reconstruction accuracy. We consider all these components below.

**Hierarchical trees enable highest forecast accuracy among time-series encoders.** Our method, `PRISM`, uses a hierarchical tree encoder to generate multiscale representations of input time series. To ensure the utility of our encoder, we have considered an alternative where we have replaced it with an MLP encoder (Table 3, "Encoder"). This alternation has degraded the performance of the model by the average of $32.94\%$. To test whether the hierarchical processing in the time domain positively affects the time-series forecasting capabilities of the model, we considered an ablation where only one layer of the tree was evaluated. We found that this ablation has degraded the performance of the model by the average of $8.83\%$, suggesting that limited temporal partitioning underexposes the model to important timescales. These results suggest the utility of hierarchical temporal encoding of data in time-series forecasting. We provide the detailed per-dataset results in Supplementary Table 6.

**Wavelets enable the highest forecast accuracy among signal feature banks.** Our model, `PRISM`, decomposes time series into hierarchical frequency components. While we chose to use wavelets for such a decomposition, `PRISM` is a general-purpose method that seamlessly admits different feature types. To evaluate whether the wavelets offer the best choice of basis functions for time-series forecasting, we have considered several classical and novel alternatives from literature (Table 3, "Frequency filter"). As a prominent new alternative, we have applied the MCD block of the D-PAD

Table 3: **Ablations and architecture search.** The **average** MSE and MAE are reported for 8 datasets over 4 seeds. We compare the performance of our method `PRISM` with different choices of encoders, frequency filters, importance scoring, loss functions, and other architecture choices. The context window is 336 for all datasets and the horizons are 96, 192, 336, 720 samples. We provide the expanded (non-averaged) data in Appendix.

| Ablation | Alternative | Mean MSE | Mean MAE | ΔMSE vs `PRISM` (%) | ΔMAE vs `PRISM` (%) |
|---|---|---|---|---|---|
| – | `PRISM` | **0.307** | **0.345** | – | – |
| Encoder | Tree-level 1 | 0.334 | 0.372 | -8.83% | -7.92% |
| | MLP | 0.408 | 0.410 | -32.94% | -18.78% |
| Frequency filter | EMA (exponential moving average) | 0.330 | 0.368 | -7.45% | -6.56% |
| | MCD (D-PAD (28) filter block) | 0.323 | 0.364 | -5.23% | -5.62% |
| | DoG (difference of Gaussians) | 0.331 | 0.369 | -7.79% | -6.92% |
| | Binomial pyramid | 0.328 | 0.367 | -6.73% | -6.41% |
| | FFT (fast Fourier transformation) | 0.351 | 0.378 | -14.32% | -9.43% |
| | Learnable filters | 0.333 | 0.370 | -8.56% | -7.14% |
| Importance scoring | All shared weights (across the entire tree) | 0.329 | 0.368 | -7.01% | -6.73% |
| | No shared weights (across the entire tree) | 0.321 | 0.364 | -4.66% | -5.46% |
| | No MLPs (all importance scores = 1) | 0.332 | 0.373 | -8.21% | -8.25% |
| | Learnable threshold (importance scores = 0 or 1) | 0.338 | 0.373 | -10.01% | -8.22% |
| Loss | No reconstruction loss | 0.333 | 0.369 | -8.46% | -6.87% |
| Architecture | No residual connections | 0.329 | 0.367 | -7.23% | -6.35% |
| | All residual connections (in every layer) | 0.337 | 0.373 | -9.80% | -8.04% |

algorithm. Among the classical choices, we considered the fast Fourier transformation (FFT), the difference of Gaussians (DoG) and the binomial pyramid. We have also tested the exponential moving average (EMA) as a simple baseline. We found that the use of the FFT has led to the substantial loss of the average time-series forecasting performance, amounting to $14.32\%$. The other classical choices, binomial pyramid, EMA, and DoG have led to a smaller yet consistent decrease of the performance amounting to $6.73\%$, $7.45\%$ and $7.79\%$ respectively. The smallest (yet still negative) gap was observed, in favor of `PRISM`, with the MCD method, amounting to $5.23\%$. We have also tried a fully learnable filter, that has led to the decrease of $-8.56\%$ Together, these results speak to the utility of the wavelets in time-series forecasting. We provide the detailed per-dataset results in Supplementary Table 7.

**Learnable importance scores per tree hierarchy level are the best way to combine wavelets.** Our method, `PRISM`, computes the importance scores for individual frequency components of the time series before adding them back together. To compute the importance scores in `PRISM`, we use a trainable MLP module. To establish the importance of the importance scoring, we performed the ablation where there is no MLP at all (Table 3, "Importance scoring"). We found that this ablation has reduced the performance of our model on our panel of datasets by the average of $8.21\%$. To try an alternative, we performed the second ablation where we replaced the MLP with a learnable threshold – similarly to wavelet literature. We found that the resulting performance was sill lagging by $10.01\%$ compared to `PRISM`.

For the MLP, our model `PRISM` uses shared weights within a tree layer but different MLP weights across the tree layers. To verify whether per-layer is the best way of the weight sharing, we trained two additional models where the MLP weights were either shared among all the tree layers or, alternatively, were not shared at all (were individual within each branch and each tree layer). Both alternations have led to the decrease of the forecasting performance on our datasets amounting to $7.01\%$ and $4.66\%$ respectively. We provide the detailed per-dataset results in Supplementary Table 8.

**Auxiliary reconstruction loss and residual connections improve time-series forecasts.** Here we have examined the impacts of our remaining design choices on the accuracy of time series forecasts. First, we have evaluated the importance of our auxiliary reconstruction loss, that has accompanied the forecast accuracy loss in `PRISM` (Table 3, "Loss"). We found that the ablation of the auxiliary reconstruction loss has decreased the overall forecast performance on our panel of datasets by the average of $8.46\%$. We then have investigated the role of the residual connections for time-series forecasting (Table 3, "Architecture"). To alter our choice of having a single set of long-range residual connections, bypassing the entire model, we considered the variants of the model architecture with residual connections in each layer and, alternatively, with no residual connections at all. We found

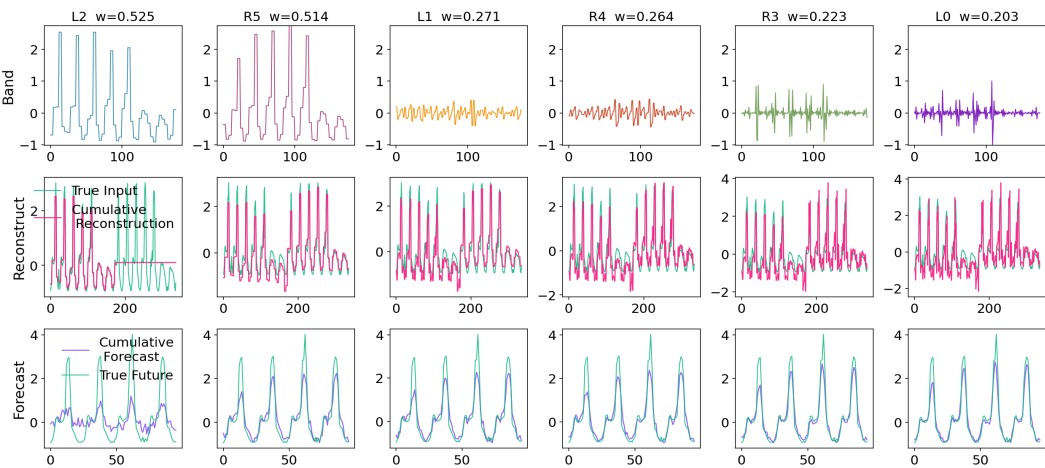

Figure 2: **Multiscale features learned by the model and their impact on time series prediction tasks.** (Top) multiscale signal components from different time- and frequency levels (L and R stands for left and right tree nodes in time decomposition; $w$ is the associated importance weight), (middle) cumulative forecast of the time series based on these components, (bottom) cumulative reconstruction of the time series from these components.

that these alternation have led to the average decreases in the model's performance amounting to $9.80\%$ and $7.23\%$ respectively. Together, these results speak toward the importance of the auxiliary reconstruction loss and long-range residual connections in time-series forecasting models. We provide the detailed per-dataset results in Supplementary Table 6.

### 4.4 TIME-FREQUENCY REPRESENTATIONS

We next sought to investigate the types of time-frequency representations that are formed at different nodes of the tree, and how they are mixed together to build forecasts. In Figure 2, we show the decomposition formed across three frequency bands (low, medium, high) in two time segments (left, right) – making the total of $2 \times 3 = 6$ representations. Although the time trees in our work could be deeper and sets of frequency features could be wider, we chose this level of decomposition for interpretability and illustration purposes. From top to bottom, in Figure 2 we show these six individual representations (top); the cumulative reconstruction of the input signal in the context window obtained as a weighted sum of these representations multiplied by their importance scores $w$ (middle; the cumulative sum goes from left representations to right representations), and the cumulative forecast for the input signal over the forecast window, obtained with the MLP head that bases on these individual representations (bottom; the cumulative sum goes from left to right).

We found that our model, PRISM, faithfully captures high-, mid-, and low-frequency components of the input signal, while also honoring the local variabilities of the signal if different temporal segments. When added, these components progressively contribute to the dynamics of time series in both reconstruction and forecasting tasks. Overall, our model successfully forecasts time series by prioritizing relevant frequency components in defined temporal domains and suppressing noise.

**Importance scores.** To evaluate whether our model, PRISM, has captured the relevant components of the data, we evaluated the importance scores—the weighting factors learned by the model to scale time-frequency components produced by the Wavelet transformation based on their importance for the forecasting in our datasets. To this end, we used the electric transformer temperature (*ETT*) datasets. These datasets offer the data collected at two different electric plants (*ETT1* and *ETT2*) collected at different sampling rates (*ETTh* at the rate of 1h and *ETTm* at the rate of 15 min). These four datasets (*ETTh1*, *ETTh2*, *ETTm1*, and *ETTm2*) provide comparable conditions with controlled variations, allowing us to evaluate whether the model learns robust and interpretable representations. To make the evaluation of our model tractable, we trained the models with the tree depth equal to 2, thus only performing one binary split of the data in time. As a reference setting, we used the *ETTm1* dataset with the context window of 336 time points, the forecast window of 720 time points, and 5 Wavelet bands.

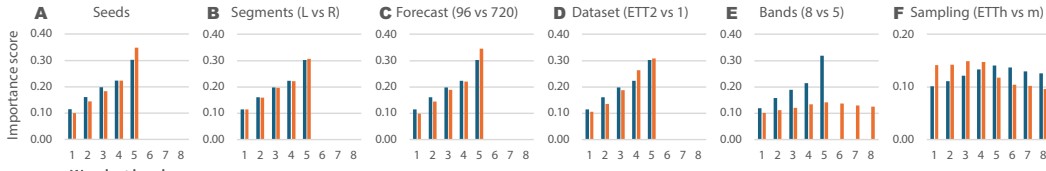

Figure 3: **Importance scores across *ETT* datasets.** The scores for the *ETTm1* dataset at forecast length of 720 (blue) compared across different (A) seeds, (B) segments of the time series, (C) forecast lengths, (D) datasets, (E) numbers of Wavelet bands, and (F) sampling rates.

For signal reconstruction, all the Wavelet components are combined with learned importance scores. Same for forecasting, we found that the importance scores learned by our model were different, increasing towards lower frequencies (Figure 3A, blue). We found that this distribution was stable across training seeds (Figure 3A), segments of the time series (left vs. right; Figure 3B), forecasting horizons (96 vs. 720 time points; Figure 3C), and datasets (*ETTm2* vs. *ETTm1*; Figure 3D) suggesting that the model has learned robust data-driven importance scores (Figure 3, blue vs. orange). Based on these results, for a detailed analysis, we averaged the importance scores across forecast horizons (96, 192, 336, and 720 time points), time series segments (left vs. right) and 2 seeds. We first evaluated the trends across different numbers of Wavelet bands (5 vs. 8 bands) and found consistent linear growth of importance scores over the first 5 bands (Figure 3E). The difference in scale is due to softmax normalization of the importance scores. Lastly, we analyzed whether the learned importance scores correspond between datasets with different sampling rates. As in the Wavelet transformation frequency doubles in every subsequent band, the fourfold difference between the sampling rates of *ETTm1* and *ETTh1* datasets (15 minutes and 1 hour respectively) corresponded to the shift of two Wavelet bands. Accordingly, we found that the sharp decrease in the importance scores of the *ETTh1* dataset came roughly 2 bands before the corresponding decrease in the *ETTm2* dataset (Figure 3F). Together, these results suggest that our model, PRISM, learns robust representations that hold across datasets, sampling rates, and hyperparameters.

## 5 DISCUSSION

We introduced PRISM, a hierarchical forecasting model that jointly organizes information across time and frequency. By recursively partitioning the input sequence and decomposing each segment into frequency bands, PRISM builds a unified representation that supports accurate and interpretable forecasts. Across standard benchmarks, this design achieves strong or state-of-the-art results while providing a transparent view of which temporal and spectral components drive predictions.

A key factor in PRISM's performance is its flexibility in selection of features from different wavelet bands. Unlike standard wavelet transforms that tie temporal and frequency resolutions, PRISM decouples them through a data-driven importance weighting mechanism. This enables the model to retain only the most predictive time–frequency components, reducing redundancy and improving generalization. Empirically, this yields compact yet expressive representations that adapt across datasets and forecast horizons.

Our analysis offers practical insights into model design. The recursive structure complements Haar wavelets' exponentially spaced frequencies and localized filters, explaining their advantage over alternative bases such as FFT. We observe that PRISM consistently prioritizes high-frequency components up to the limit imposed by the forecast window and converges to a small number of informative bands. These patterns suggest simple heuristics for initialization—e.g., using roughly $\log_2 T$ frequency levels and two temporal splits for common forecasting setups.

Looking ahead, several extensions are promising. Learning adaptive frequency bases or irregular temporal partitions could further enhance flexibility, and extending the framework to multivariate or cross-domain forecasting would test its scalability. More broadly, PRISM exemplifies how combining analytical structure with data-driven selection can produce models that are accurate, interpretable, and efficient.

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

# APPENDIX

## A  SUPPLEMENTARY DISCUSSION

### A.1  MODEL DESIGN

The performance of our model is enabled by the architectural pairing with the Haar wavelets. Haar wavelets are hierarchical filters that decompose time series into components of different frequencies sampled at different times (Supplementary Figure 4 left). The frequency filters start from the highest frequency afforded by the discrete digital data. Each subsequent filter is tuned to a frequency that is one half of the previous one, thus selecting exponentially spaced frequencies. Similarly, the Haar's wavelets provide a hierarchical organization in the time domain. While the initial highest-frequency filters are individually applied to the smallest time segments of the data, each subsequent filter is applied to segments twice as large. This way, the joint time-frequency decomposition with the Haar's wavelets always stays at the forefront of the time-frequency resolution.

While such a decomposition provides the most detailed view of the data, it creates an exponentially large number of time-frequency segments, many of which may be too detailed of irrelevant for the time series forecasting. This has motivated our method as follows. Compared to the Haar's wavelets, our method, PRISM, has a complementary structure where time series are progressively split, through bisection, into smaller segments where each following segment is twice as short as the previous one. On each time decomposition level, the full Wavelet decomposition is applied to each temporal segment of the data, and the output Wavelet-generated components corresponding to the same frequency are concatenated in time. The process is iterated across all the levels of temporal decomposition of the original time series, thus producing all possible frequency components at all possible time decomposition levels. Thus, unlike in the original Haar's wavelets, the frequency resolution is no longer tied to the temporal resolution.

So far, such a procedure generates the number of the components that is even larger than that in the original Haar's wavelet transformation. To account for that fact, our method, PRISM, then only selects the time-frequency levels that are important for forecasting through a data-driven importance scoring procedure. In the limit, the model may fully suppress irrelevant time-frequency representations, thus going below the number of the components in the original Haar's wavelet transformation (Figure 4 right). This way our approach, PRISM, through the architectural pairing with the Haar's wavelets, inherits its good properties and furthers its performance though data-driven post-processing.

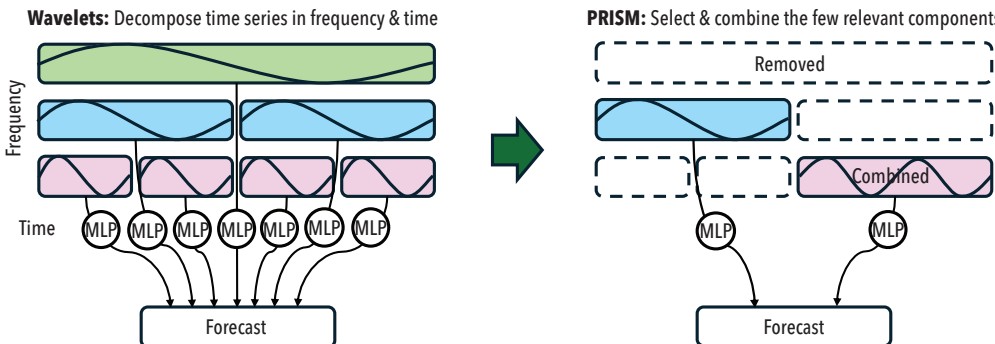

Figure 4: **PRISM interpretation scheme.** (Left) The Wavelet transformation decomposes time series into components localized in frequency and time. The time series forecast then can be done using feedforward MLP networks applied to each Wavelet component. The required number of MLPs grows exponentially with the number of Wavelet levels. (Right) PRISM effectively reduces the number of required MLPs by selecting only the Wavelet components relevant for forecasting and combining them to the extents necessary for forecasting.

This parallel between the architecture of Haar's wavelets and the architecture of our method, PRISM, provides the insights into the model's optimal design choices, hyperparameters, and its learned importance scores. First, it explains why our model has achieved the best results when paired with Haar's wavelets (Table 3; Supplementary Table 7): Other frequency features, including the FFT, have led to lesser results when paired with PRISM because they use different frequency banks

corresponding, explicitly or implicitly through the Gabor's uncertainty principle, to different levels of temporal abstraction that are not naturally paired with the binary tree structure, used here in `PRISM`. For the Haar's wavelets, our model, `PRISM`, offers an improvement that can be explained by selecting from the (otherwise exponential) number of time-frequency components and sombining them into a smaller number of task-relevant components. This allows for data-driven time-frequency tradeoff that may improve the model's forecasting performance by effectively reducing its number of parameters and thus making it less prone to overfitting while keeping the flexibility to capture the relevant dimensions of the data.

## A.2 MODEL HYPERPARAMETERS

Second, the joint consideration of the Haar's wavelets and `PRISM` helps understanding the time-frequency segments selected by the model and thus inform the initial hyperparameter selection for new data and tasks. Unlike many other filter banks, Haar's wavelets start with extracting the highest frequency afforded by the data, then progressively recruiting lower frequencies but not necessarily requiring them. This design choice is substantiated because the sampling rates of time series data are typically chosen such that the data's finest resolution still contains useful information (and everything beyond it, being noise, is filtered out to save disk space). Accordingly, across the datasets in our experiments, the highest-frequency component has always received a high importance score from the model (Figure 3). On the other end of the spectrum, the lowest frequency that may be relevant for the time series forecasting is inversely proportional to the context window length $T$ through Gabor's uncertainty principle. Considered together with the exponential spacing of Haar's wavelet frequencies, that suggests that the number of Haar's wavelets should not exceed $\log_2 T$. In the example of a context window of $T = 336$ time points considered in this work, the number of filters thus should not exceed $\log_2 336 \approx 8$. Aligned with this reasoning, we observed that, for the context window of $T = 336$ time points, 6 Wavelet components were consistently receiving high importance scores from `PRISM`, while subsequent filters got consistently suppressed (Figure 3). This corresponds to $\tilde{5}$ filters leading to the model's best performance in out hyperparameter search (Supplementary Table 9). Lastly, for the time domain decomposition, we have experimentally found that 2 levels of the temporal split were optimal across the tasks (Supplementary Table 10). This may be explained by the model tracking the trend of the data through the differences between early and late time points. Overall, these considerations suggest that, for new datasets, a good initial performance should be expected with using Haar's filters up to the Gabor's limit over 2 temporal segments.

## A.3 COMPARISON OF TRAINING TIMES

Besides the forecasting accuracy, another important property of time-series models is the training time. To evaluate how our model compares to other prominent models in terms of the runtime, we evaluated the training speed of several models on the ETTh1 dataset using the forecast window of 96 timepoints, one of the most popular evaluation settings among the datasets considered in this work. We followed the standard approach and recorded the times that were necessary for the training of multiple (ten) epochs to negate the effects of the training overhead. For our model, `PRISM`, the training of 10 epochs took 65 seconds. We found that, this way, our model has shown the two-fold improvement over its closest counterpart, the D-PAD, whose training took 141 seconds for 10 epochs. The next counterpart of our model, DLinear, was somewhat faster to train, taking 42 seconds per 10 epochs, however, this speedup came at the cost of the decreased average performance in the forecasting task (-2.95% MSE). Continuing this trend, iTransformer took 32 second per 10 epochs at the performance drop of 4.70% MSE. The fastest results were shown by PatchTST, N-HiTS, and xPatch with 12, 10, and 7 seconds per 10 epochs respectively, at the performance decrease of 1.70, 6.87, and 3.45% respectively. Overall, these results suggest that our model offers a reasonable time-accuracy tradeoff while PatchTST also fares well on this metric.

Table 4: Extended results (GIFT dataset).

| Dataset | PRISM | | D-PAD | | DLinear | |
|---|---|---|---|---|---|---|
| | MSE | MAE | MSE | MAE | MSE | MAE |
| bitbrains_fast_storage/5T/medium | 4.837E+06 | 6.209E+02 | 6.142E+06 | 7.046E+02 | 2.378E+06 | 2.948E+02 |
| bitbrains_fast_storage/5T/short | 7.175E+06 | 6.355E+02 | 6.720E+06 | 6.079E+02 | 7.843E+06 | 6.304E+02 |
| bitbrains_fast_storage/H/short | 3.551E+06 | 6.090E+02 | 3.656E+06 | 6.043E+02 | 7.197E+06 | 7.899E+02 |
| bitbrains_rnd/5T/long | 5.124E+06 | 5.592E+02 | 5.213E+06 | 5.612E+02 | 4.037E+06 | 3.744E+02 |
| bitbrains_rnd/5T/medium | 1.104E+06 | 3.280E+02 | 2.007E+06 | 3.846E+02 | 3.548E+05 | 1.425E+02 |
| bitbrains_rnd/5T/short | 3.758E+06 | 4.406E+02 | 3.896E+06 | 4.167E+02 | 4.223E+06 | 4.155E+02 |
| bitbrains_rnd/H/short | 1.625E+06 | 3.244E+02 | 1.526E+06 | 3.000E+02 | 4.532E+06 | 5.015E+02 |
| bizitobs_application/10S/long | 1.186E+09 | 2.557E+04 | 1.201E+09 | 2.557E+04 | 1.201E+09 | 2.557E+04 |
| bizitobs_application/10S/medium | 1.229E+09 | 2.638E+04 | 1.255E+09 | 2.638E+04 | 1.333E+09 | 2.639E+04 |
| bizitobs_application/10S/short | 1.273E+09 | 2.676E+04 | 1.330E+09 | 2.704E+04 | 1.365E+09 | 2.719E+04 |
| bizitobs_l2c/5T/long | 4.595E+02 | 1.708E+01 | 5.236E+02 | 1.781E+01 | 5.656E+02 | 1.850E+01 |
| bizitobs_l2c/5T/medium | 3.457E+02 | 1.483E+01 | 4.083E+02 | 1.568E+01 | 4.660E+02 | 1.671E+01 |
| bizitobs_l2c/5T/short | 5.768E+02 | 1.942E+01 | 5.976E+02 | 1.979E+01 | 6.286E+02 | 2.027E+01 |
| bizitobs_l2c/H/medium | 1.628E+02 | 1.148E+01 | 4.099E+02 | 1.838E+01 | 2.538E+02 | 1.452E+01 |
| bizitobs_l2c/H/short | 2.873E+02 | 1.402E+01 | 4.381E+02 | 1.650E+01 | 3.896E+02 | 1.556E+01 |
| bizitobs_service/10S/long | 9.151E+06 | 1.869E+03 | 9.761E+06 | 1.939E+03 | 9.762E+06 | 1.939E+03 |
| bizitobs_service/10S/medium | 9.645E+06 | 1.929E+03 | 1.023E+07 | 1.993E+03 | 1.045E+07 | 2.015E+03 |
| bizitobs_service/10S/short | 1.012E+07 | 1.996E+03 | 6.792E+06 | 1.347E+03 | 1.033E+07 | 2.019E+03 |
| car_parts/M/short | 1.560E+00 | 6.070E-01 | 1.590E+00 | 6.560E-01 | 1.610E+00 | 5.780E-01 |
| covid_deaths/D/short | 1.322E+08 | 2.676E+03 | 2.049E+08 | 3.709E+03 | 3.294E+08 | 4.596E+03 |
| electricity/15T/medium | 2.941E+06 | 4.837E+02 | 6.769E+06 | 6.565E+02 | 4.842E+06 | 5.816E+02 |
| electricity/15T/short | 4.269E+06 | 5.806E+02 | 3.344E+06 | 5.271E+02 | 4.493E+06 | 5.936E+02 |
| electricity/D/short | 1.171E+11 | 7.898E+04 | 1.077E+11 | 5.912E+04 | 1.172E+11 | 7.951E+04 |
| electricity/H/long | 1.218E+08 | 2.709E+03 | 1.346E+08 | 2.795E+03 | 1.248E+08 | 2.727E+03 |
| electricity/H/medium | 1.090E+08 | 2.642E+03 | 1.667E+08 | 2.961E+03 | 2.404E+08 | 3.319E+03 |
| electricity/H/short | 4.011E+07 | 1.800E+03 | 3.281E+07 | 1.652E+03 | 4.335E+07 | 1.863E+03 |
| electricity/W/short | 1.568E+12 | 3.538E+05 | 1.590E+12 | 3.138E+05 | 1.570E+12 | 3.557E+05 |
| ett1/15T/long | 2.637E+01 | 4.060E+00 | 2.879E+01 | 4.220E+00 | 2.529E+01 | 3.982E+00 |
| ett1/15T/medium | 2.467E+01 | 3.980E+00 | 5.113E+01 | 5.564E+00 | 2.880E+01 | 4.220E+00 |
| ett1/15T/short | 2.034E+01 | 3.540E+00 | 2.615E+01 | 4.060E+00 | 2.232E+01 | 3.756E+00 |
| ett1/D/short | 1.396E+05 | 2.741E+02 | 1.563E+05 | 2.943E+02 | 1.470E+05 | 2.864E+02 |
| ett1/H/long | 5.245E+02 | 1.819E+01 | 6.983E+02 | 2.055E+01 | 9.406E+02 | 2.420E+01 |
| ett1/H/medium | 4.540E+02 | 1.641E+01 | 5.027E+02 | 1.788E+01 | 6.762E+02 | 2.051E+01 |
| ett1/H/short | 3.568E+02 | 1.472E+01 | 3.635E+02 | 1.411E+01 | 4.647E+02 | 1.692E+01 |
| ett1/W/short | 7.552E+06 | 2.099E+03 | 8.856E+06 | 2.246E+03 | 1.051E+07 | 2.337E+03 |
| ett2/15T/long | 6.015E+02 | 1.976E+01 | 6.083E+02 | 1.987E+01 | 6.077E+02 | 1.988E+01 |
| ett2/15T/medium | 6.243E+02 | 2.021E+01 | 6.396E+02 | 2.043E+01 | 6.575E+02 | 2.068E+01 |
| ett2/15T/short | 7.225E+02 | 2.162E+01 | 7.189E+02 | 2.158E+01 | 7.347E+02 | 2.179E+01 |
| ett2/D/short | 5.056E+06 | 1.817E+03 | 5.062E+06 | 1.817E+03 | 5.204E+06 | 1.841E+03 |
| ett2/H/long | 9.011E+03 | 7.706E+01 | 9.209E+03 | 7.812E+01 | 9.663E+03 | 7.940E+01 |
| ett2/H/medium | 1.051E+04 | 8.351E+01 | 1.083E+04 | 8.436E+01 | 1.124E+04 | 8.560E+01 |
| ett2/H/short | 1.117E+04 | 8.519E+01 | 9.934E+03 | 8.007E+01 | 1.150E+04 | 8.627E+01 |
| ett2/W/short | 2.374E+08 | 1.244E+04 | 2.526E+08 | 1.286E+04 | 2.757E+08 | 1.339E+04 |
| hierarchical_sales/D/short | 4.600E+01 | 3.720E+00 | 4.671E+01 | 3.730E+00 | 4.897E+01 | 3.550E+00 |
| hierarchical_sales/W/short | 1.304E+03 | 2.184E+01 | 1.236E+03 | 2.045E+01 | 1.571E+03 | 2.390E+01 |
| hospital/M/short | 1.537E+06 | 4.545E+02 | 1.544E+06 | 4.555E+02 | 1.544E+06 | 4.554E+02 |
| jena_weather/10T/long | 2.212E+05 | 2.575E+02 | 2.223E+05 | 2.583E+02 | 2.250E+05 | 2.608E+02 |
| jena_weather/10T/medium | 2.184E+05 | 2.600E+02 | 2.187E+05 | 2.591E+02 | 2.198E+05 | 2.618E+02 |
| jena_weather/10T/short | 2.167E+05 | 2.569E+02 | 2.115E+05 | 2.551E+02 | 2.171E+05 | 2.576E+02 |
| jena_weather/D/short | 2.057E+05 | 2.652E+02 | 2.069E+05 | 2.679E+02 | 2.092E+05 | 2.765E+02 |
| jena_weather/H/long | 2.065E+05 | 2.533E+02 | 2.087E+05 | 2.672E+02 | 2.146E+05 | 2.668E+02 |
| jena_weather/H/medium | 2.150E+05 | 2.614E+02 | 2.160E+05 | 2.574E+02 | 2.197E+05 | 2.623E+02 |
| jena_weather/H/short | 2.209E+05 | 2.581E+02 | 1.508E+05 | 2.013E+02 | 2.220E+05 | 2.592E+02 |
| kdd_cup_2018/D/short | 2.705E+03 | 3.847E+01 | 2.947E+03 | 4.060E+01 | 3.104E+03 | 4.158E+01 |
| kdd_cup_2018/H/long | 1.418E+03 | 2.885E+01 | 1.814E+03 | 3.229E+01 | 1.805E+03 | 2.994E+01 |
| kdd_cup_2018/H/medium | 3.324E+03 | 4.185E+01 | 3.194E+03 | 4.022E+01 | 4.313E+03 | 4.369E+01 |
| kdd_cup_2018/H/short | 6.413E+03 | 5.371E+01 | 5.954E+03 | 5.123E+01 | 5.950E+03 | 4.894E+01 |
| loop_seattle/5T/medium | 4.988E+02 | 1.559E+01 | 3.513E+02 | 1.245E+01 | 5.541E+02 | 1.644E+01 |
| loop_seattle/5T/short | 2.194E+02 | 9.350E+00 | 2.165E+02 | 9.260E+00 | 2.261E+02 | 9.439E+00 |
| loop_seattle/D/short | 4.046E+01 | 4.990E+00 | 2.285E+03 | 4.759E+01 | 5.891E+01 | 6.196E+00 |
| loop_seattle/H/long | 1.547E+01 | 3.040E+00 | 1.554E+01 | 3.080E+00 | 1.863E+01 | 3.347E+00 |
| loop_seattle/H/medium | 1.233E+01 | 2.730E+00 | 4.488E+01 | 5.460E+00 | 1.340E+01 | 2.888E+00 |
| loop_seattle/H/short | 1.509E+01 | 3.000E+00 | 1.404E+01 | 2.930E+00 | 1.560E+01 | 3.049E+00 |

Table 5: Extended results (GIFT dataset)—continued.

| Dataset | PRISM | | D-PAD | | DLinear | |
|---|---|---|---|---|---|---|
| | MSE | MAE | MSE | MAE | MSE | MAE |
| m4_daily/D/short | 5.645E+07 | 5.801E+03 | 5.472E+07 | 5.718E+03 | 5.661E+07 | 5.808E+03 |
| m4_hourly/H/short | 3.259E+09 | 1.315E+04 | 3.348E+09 | 1.330E+04 | 3.557E+09 | 1.375E+04 |
| m4_monthly/M/short | 2.534E+07 | 3.679E+03 | 2.421E+07 | 3.608E+03 | 2.589E+07 | 3.704E+03 |
| m4_quarterly/Q/short | 3.670E+07 | 4.556E+03 | 3.697E+07 | 4.570E+03 | 3.696E+07 | 4.577E+03 |
| m4_weekly/W/short | 5.422E+07 | 4.999E+03 | 5.507E+07 | 5.042E+03 | 5.525E+07 | 5.044E+03 |
| m4_yearly/A/short | 2.915E+07 | 3.971E+03 | 2.931E+07 | 3.971E+03 | 2.920E+07 | 3.969E+03 |
| m_dense/D/short | 6.907E+05 | 5.554E+02 | 7.493E+05 | 5.796E+02 | 7.296E+05 | 5.721E+02 |
| m_dense/H/long | 2.195E+05 | 3.031E+02 | 2.404E+05 | 3.183E+02 | 2.364E+05 | 3.110E+02 |
| m_dense/H/medium | 2.511E+05 | 3.167E+02 | 2.995E+05 | 3.460E+02 | 4.640E+05 | 4.286E+02 |
| m_dense/H/short | 2.096E+05 | 2.901E+02 | 2.072E+05 | 2.976E+02 | 2.125E+05 | 2.899E+02 |
| restaurant/D/short | 5.175E+02 | 1.707E+01 | 5.033E+02 | 1.677E+01 | 1.006E+03 | 2.565E+01 |
| saugeen/D/short | 8.933E+02 | 2.142E+01 | 1.491E+03 | 2.722E+01 | 1.264E+03 | 2.507E+01 |
| saugeen/M/short | 2.111E+02 | 1.113E+01 | 3.179E+02 | 1.653E+01 | 2.278E+02 | 1.200E+01 |
| saugeen/W/short | 1.771E+03 | 2.474E+01 | 3.099E+03 | 3.341E+01 | 2.109E+03 | 2.788E+01 |
| solar/10T/long | 2.227E-01 | 3.295E-01 | 3.210E-01 | 3.780E-01 | 1.900E-05 | 3.000E-03 |
| solar/10T/medium | 6.024E+01 | 5.300E+00 | 6.320E+01 | 5.382E+00 | 6.868E+01 | 5.541E+00 |
| solar/10T/short | 3.813E+01 | 4.000E+00 | 6.131E+01 | 5.550E+00 | 3.687E+01 | 3.850E+00 |
| solar/D/short | 2.718E+05 | 3.492E+02 | 5.787E+05 | 6.039E+02 | 2.819E+05 | 3.549E+02 |
| solar/H/long | 1.346E+01 | 2.770E+00 | 7.272E+00 | 2.120E+00 | 9.700E-02 | 2.090E-01 |
| solar/H/medium | 3.930E+00 | 1.540E+00 | 4.178E+01 | 5.852E+00 | 1.272E-02 | 1.019E-01 |
| solar/H/short | 8.700E-02 | 2.320E-01 | 6.857E+03 | 7.672E+01 | 4.749E+03 | 6.048E+01 |
| solar/W/short | 1.454E+07 | 2.697E+03 | 3.313E+07 | 5.320E+03 | 1.598E+07 | 2.883E+03 |
| sz_taxi/15T/medium | 1.665E+02 | 9.670E+00 | 1.669E+02 | 9.673E+00 | 1.525E+02 | 9.244E+00 |
| sz_taxi/15T/short | 1.474E+02 | 9.220E+00 | 1.391E+02 | 8.940E+00 | 1.547E+02 | 9.446E+00 |
| sz_taxi/H/short | 1.701E+02 | 9.840E+00 | 1.776E+02 | 1.011E+01 | 1.717E+02 | 9.941E+00 |
| temperature_rain/D/short | 4.836E+02 | 1.311E+01 | 4.371E+02 | 1.178E+01 | 4.553E+02 | 1.252E+01 |
| us_births/D/short | 1.375E+06 | 9.429E+02 | 2.688E+06 | 1.255E+03 | 2.231E+06 | 1.167E+03 |
| us_births/M/short | 8.784E+06 | 2.583E+03 | 4.851E+10 | 1.570E+05 | 2.342E+07 | 4.037E+03 |
| us_births/W/short | 1.298E+07 | 2.733E+03 | 2.050E+07 | 3.558E+03 | 1.724E+07 | 3.256E+03 |
| *Best count (MSE)* | 61 | – | 21 | – | 9 | – |
| *Best count (MAE)* | – | 52 | – | 25 | – | 14 |

Table 6: Extended ablation results (encoder, loss, and architecture ablations).

| Dataset | Context | H | PRISM | | Tree-level 1 | | MLP | | No recon loss | | No residuals | | All residuals | |
|---|---|---|---|---|---|---|---|---|---|---|---|---|---|---|
| | | | MSE | MAE | MSE | MAE | MSE | MAE | MSE | MAE | MSE | MAE | MSE | MAE |
| ETTh1 | 336 | 96 | 0.355 | 0.374 | 0.373 | 0.401 | 0.368 | 0.388 | 0.375 | 0.402 | 0.361 | 0.392 | 0.387 | 0.409 |
| ETTh1 | 336 | 192 | 0.390 | 0.405 | 0.413 | 0.422 | 0.409 | 0.418 | 0.425 | 0.431 | 0.409 | 0.420 | 0.434 | 0.436 |
| ETTh1 | 336 | 336 | 0.386 | 0.412 | 0.413 | 0.435 | 0.404 | 0.424 | 0.429 | 0.442 | 0.406 | 0.427 | 0.438 | 0.446 |
| ETTh1 | 336 | 720 | 0.421 | 0.445 | 0.466 | 0.476 | 0.449 | 0.461 | 0.464 | 0.474 | 0.464 | 0.475 | 0.514 | 0.502 |
| ETTh2 | 336 | 96 | 0.267 | 0.322 | 0.273 | 0.335 | 0.281 | 0.341 | 0.278 | 0.343 | 0.267 | 0.331 | 0.282 | 0.345 |
| ETTh2 | 336 | 192 | 0.311 | 0.359 | 0.325 | 0.370 | 0.332 | 0.376 | 0.338 | 0.382 | 0.324 | 0.370 | 0.330 | 0.381 |
| ETTh2 | 336 | 336 | 0.318 | 0.364 | 0.335 | 0.385 | 0.338 | 0.385 | 0.337 | 0.393 | 0.325 | 0.382 | 0.339 | 0.391 |
| ETTh2 | 336 | 720 | 0.390 | 0.421 | 0.403 | 0.437 | 0.408 | 0.435 | 0.414 | 0.443 | 0.401 | 0.435 | 0.410 | 0.443 |
| ETTm1 | 336 | 96 | 0.288 | 0.321 | 0.293 | 0.347 | 0.292 | 0.347 | 0.295 | 0.349 | 0.294 | 0.345 | 0.303 | 0.356 |
| ETTm1 | 336 | 192 | 0.325 | 0.359 | 0.335 | 0.369 | 0.341 | 0.372 | 0.333 | 0.371 | 0.332 | 0.370 | 0.340 | 0.376 |
| ETTm1 | 336 | 336 | 0.358 | 0.379 | 0.363 | 0.389 | 0.371 | 0.391 | 0.367 | 0.392 | 0.364 | 0.390 | 0.375 | 0.398 |
| ETTm1 | 336 | 720 | 0.410 | 0.419 | 0.426 | 0.423 | 0.428 | 0.426 | 0.431 | 0.425 | 0.427 | 0.423 | 0.433 | 0.430 |
| ETTm2 | 336 | 96 | 0.158 | 0.234 | 0.182 | 0.272 | 0.177 | 0.263 | 0.167 | 0.245 | 0.169 | 0.260 | 0.176 | 0.268 |
| ETTm2 | 336 | 192 | 0.214 | 0.267 | 0.258 | 0.375 | 0.234 | 0.305 | 0.232 | 0.305 | 0.227 | 0.300 | 0.225 | 0.299 |
| ETTm2 | 336 | 336 | 0.264 | 0.327 | 0.288 | 0.340 | 0.291 | 0.339 | 0.285 | 0.332 | 0.280 | 0.334 | 0.294 | 0.343 |
| ETTm2 | 336 | 720 | 0.360 | 0.383 | 0.361 | 0.385 | 0.364 | 0.386 | 0.370 | 0.396 | 0.361 | 0.386 | 0.370 | 0.392 |
| Exchange | 336 | 96 | 0.081 | 0.198 | 0.088 | 0.210 | 0.186 | 0.320 | 0.086 | 0.209 | 0.088 | 0.213 | 0.087 | 0.210 |
| Exchange | 336 | 192 | 0.176 | 0.297 | 0.177 | 0.303 | 0.544 | 0.506 | 0.179 | 0.304 | 0.188 | 0.315 | 0.180 | 0.303 |
| Exchange | 336 | 336 | 0.314 | 0.396 | 0.344 | 0.427 | 0.998 | 0.775 | 0.347 | 0.429 | 0.341 | 0.424 | 0.336 | 0.427 |
| Exchange | 336 | 720 | 0.712 | 0.606 | 0.968 | 0.756 | 1.673 | 1.027 | 0.905 | 0.701 | 0.939 | 0.734 | 0.901 | 0.709 |
| Weather | 336 | 96 | 0.140 | 0.177 | 0.157 | 0.206 | 0.155 | 0.220 | 0.158 | 0.208 | 0.156 | 0.205 | 0.159 | 0.210 |
| Weather | 336 | 192 | 0.187 | 0.227 | 0.201 | 0.248 | 0.205 | 0.255 | 0.202 | 0.248 | 0.202 | 0.249 | 0.202 | 0.249 |
| Weather | 336 | 336 | 0.234 | 0.266 | 0.252 | 0.286 | 0.260 | 0.293 | 0.252 | 0.288 | 0.252 | 0.288 | 0.253 | 0.287 |
| Weather | 336 | 720 | 0.302 | 0.311 | 0.324 | 0.339 | 0.327 | 0.343 | 0.323 | 0.339 | 0.326 | 0.339 | 0.323 | 0.336 |
| Mean | | | 0.307 | 0.345 | 0.334 | 0.372 | 0.408 | 0.410 | 0.333 | 0.369 | 0.329 | 0.367 | 0.337 | 0.373 |
| Difference | | | | | -8.83% | -7.92% | -32.94% | -18.78% | -8.46% | -6.87% | -7.23% | -6.35% | -9.80% | -8.04% |

Table 7: Extended ablation results (frequency filter ablations).

| Dataset | Context | H | PRISM | | Binomial pyramid | | FFT | | EMA | | MCD | | DoG | | Learnable filters | |
|---|---|---|---|---|---|---|---|---|---|---|---|---|---|---|---|---|
| | | | MSE | MAE | MSE | MAE | MSE | MAE | MSE | MAE | MSE | MAE | MSE | MAE | MSE | MAE |
| ETTh1 | 336 | 96 | 0.355 | 0.374 | 0.369 | 0.395 | 0.383 | 0.406 | 0.375 | 0.398 | 0.377 | 0.403 | 0.380 | 0.406 | 0.373 | 0.397 |
| ETTh1 | 336 | 192 | 0.390 | 0.405 | 0.413 | 0.422 | 0.425 | 0.431 | 0.402 | 0.412 | 0.416 | 0.424 | 0.424 | 0.429 | 0.420 | 0.425 |
| ETTh1 | 336 | 336 | 0.386 | 0.412 | 0.411 | 0.432 | 0.425 | 0.441 | 0.397 | 0.422 | 0.392 | 0.418 | 0.427 | 0.439 | 0.413 | 0.433 |
| ETTh1 | 336 | 720 | 0.421 | 0.445 | 0.473 | 0.479 | 0.944 | 0.681 | 0.467 | 0.472 | 0.453 | 0.461 | 0.490 | 0.491 | 0.466 | 0.475 |
| ETTh2 | 336 | 96 | 0.267 | 0.322 | 0.281 | 0.344 | 0.270 | 0.331 | 0.274 | 0.339 | 0.275 | 0.338 | 0.282 | 0.343 | 0.273 | 0.338 |
| ETTh2 | 336 | 192 | 0.311 | 0.359 | 0.335 | 0.380 | 0.360 | 0.391 | 0.331 | 0.377 | 0.345 | 0.390 | 0.336 | 0.380 | 0.327 | 0.369 |
| ETTh2 | 336 | 336 | 0.318 | 0.364 | 0.337 | 0.393 | 0.336 | 0.392 | 0.314 | 0.373 | 0.317 | 0.376 | 0.337 | 0.391 | 0.325 | 0.382 |
| ETTh2 | 336 | 720 | 0.390 | 0.421 | 0.413 | 0.445 | 0.407 | 0.441 | 0.390 | 0.428 | 0.399 | 0.437 | 0.406 | 0.437 | 0.406 | 0.439 |
| ETTm1 | 336 | 96 | 0.288 | 0.321 | 0.302 | 0.358 | 0.295 | 0.348 | 0.289 | 0.345 | 0.298 | 0.353 | 0.296 | 0.349 | 0.303 | 0.356 |
| ETTm1 | 336 | 192 | 0.325 | 0.359 | 0.338 | 0.374 | 0.333 | 0.369 | 0.329 | 0.367 | 0.334 | 0.374 | 0.334 | 0.365 | 0.343 | 0.377 |
| ETTm1 | 336 | 336 | 0.358 | 0.379 | 0.371 | 0.396 | 0.365 | 0.388 | 0.364 | 0.389 | 0.366 | 0.392 | 0.369 | 0.393 | 0.364 | 0.390 |
| ETTm1 | 336 | 720 | 0.410 | 0.419 | 0.430 | 0.427 | 0.435 | 0.432 | 0.423 | 0.423 | 0.420 | 0.425 | 0.426 | 0.423 | 0.426 | 0.422 |
| ETTm2 | 336 | 96 | 0.158 | 0.234 | 0.176 | 0.265 | 0.177 | 0.270 | 0.172 | 0.262 | 0.173 | 0.263 | 0.172 | 0.263 | 0.171 | 0.261 |
| ETTm2 | 336 | 192 | 0.214 | 0.267 | 0.235 | 0.306 | 0.235 | 0.306 | 0.227 | 0.301 | 0.233 | 0.306 | 0.232 | 0.304 | 0.234 | 0.307 |
| ETTm2 | 336 | 336 | 0.264 | 0.327 | 0.291 | 0.342 | 0.291 | 0.344 | 0.283 | 0.338 | 0.286 | 0.341 | 0.285 | 0.340 | 0.287 | 0.339 |
| ETTm2 | 336 | 720 | 0.360 | 0.383 | 0.368 | 0.391 | 0.369 | 0.393 | 0.361 | 0.385 | 0.363 | 0.387 | 0.364 | 0.390 | 0.363 | 0.387 |
| Exchange | 336 | 96 | 0.081 | 0.198 | 0.094 | 0.215 | 0.093 | 0.217 | 0.092 | 0.215 | 0.093 | 0.217 | 0.097 | 0.224 | 0.099 | 0.223 |
| Exchange | 336 | 192 | 0.176 | 0.297 | 0.179 | 0.307 | 0.217 | 0.335 | 0.187 | 0.311 | 0.180 | 0.308 | 0.212 | 0.331 | 0.195 | 0.322 |
| Exchange | 336 | 336 | 0.314 | 0.396 | 0.355 | 0.436 | 0.336 | 0.421 | 0.507 | 0.541 | 0.323 | 0.410 | 0.367 | 0.445 | 0.329 | 0.421 |
| Exchange | 336 | 720 | 0.712 | 0.606 | 0.754 | 0.616 | 0.787 | 0.633 | 0.792 | 0.634 | 0.784 | 0.651 | 0.759 | 0.620 | 0.973 | 0.753 |
| Weather | 336 | 96 | 0.140 | 0.177 | 0.163 | 0.212 | 0.162 | 0.212 | 0.161 | 0.210 | 0.155 | 0.206 | 0.166 | 0.216 | 0.152 | 0.202 |
| Weather | 336 | 192 | 0.187 | 0.227 | 0.207 | 0.252 | 0.208 | 0.254 | 0.212 | 0.260 | 0.196 | 0.243 | 0.203 | 0.250 | 0.197 | 0.242 |
| Weather | 336 | 336 | 0.234 | 0.266 | 0.245 | 0.286 | 0.246 | 0.287 | 0.245 | 0.285 | 0.252 | 0.286 | 0.246 | 0.287 | 0.247 | 0.281 |
| Weather | 336 | 720 | 0.302 | 0.311 | 0.326 | 0.339 | 0.325 | 0.339 | 0.323 | 0.336 | 0.323 | 0.336 | 0.333 | 0.337 | 0.314 | 0.330 |
| Mean | | | 0.307 | 0.345 | 0.328 | 0.367 | 0.351 | 0.378 | 0.330 | 0.368 | 0.323 | 0.364 | 0.331 | 0.369 | 0.333 | 0.370 |
| Difference | | | | | -6.73% | -6.41% | -14.32% | -9.43% | -7.45% | -6.56% | -5.23% | -5.62% | -7.79% | -6.92% | -8.56% | -7.14% |

Table 8: Extended ablation results (importance scoring ablations).

| Dataset | Context | H | PRISM | | No MLP | | No shared weights | | All shared weights | | Learned thresholds | |
|---|---|---|---|---|---|---|---|---|---|---|---|---|
| | | | MSE | MAE | MSE | MAE | MSE | MAE | MSE | MAE | MSE | MAE |
| ETTh1 | 336 | 96 | 0.355 | 0.374 | 0.384 | 0.414 | 0.363 | 0.393 | 0.361 | 0.392 | 0.501 | 0.476 |
| ETTh1 | 336 | 192 | 0.390 | 0.405 | 0.421 | 0.432 | 0.409 | 0.419 | 0.408 | 0.419 | 0.422 | 0.433 |
| ETTh1 | 336 | 336 | 0.386 | 0.412 | 0.420 | 0.441 | 0.405 | 0.427 | 0.408 | 0.430 | 0.420 | 0.441 |
| ETTh1 | 336 | 720 | 0.421 | 0.445 | 0.466 | 0.475 | 0.465 | 0.474 | 0.466 | 0.474 | 0.466 | 0.475 |
| ETTh2 | 336 | 96 | 0.267 | 0.322 | 0.275 | 0.337 | 0.267 | 0.331 | 0.268 | 0.331 | 0.301 | 0.359 |
| ETTh2 | 336 | 192 | 0.311 | 0.359 | 0.330 | 0.371 | 0.324 | 0.380 | 0.326 | 0.370 | 0.327 | 0.372 |
| ETTh2 | 336 | 336 | 0.318 | 0.364 | 0.328 | 0.382 | 0.335 | 0.381 | 0.325 | 0.381 | 0.328 | 0.382 |
| ETTh2 | 336 | 720 | 0.390 | 0.421 | 0.404 | 0.437 | 0.407 | 0.441 | 0.401 | 0.434 | 0.401 | 0.435 |
| ETTm1 | 336 | 96 | 0.288 | 0.321 | 0.299 | 0.352 | 0.295 | 0.349 | 0.294 | 0.348 | 0.314 | 0.364 |
| ETTm1 | 336 | 192 | 0.325 | 0.359 | 0.336 | 0.373 | 0.331 | 0.369 | 0.339 | 0.369 | 0.335 | 0.373 |
| ETTm1 | 336 | 336 | 0.358 | 0.379 | 0.367 | 0.393 | 0.364 | 0.389 | 0.366 | 0.389 | 0.367 | 0.393 |
| ETTm1 | 336 | 720 | 0.410 | 0.419 | 0.428 | 0.425 | 0.428 | 0.422 | 0.426 | 0.422 | 0.429 | 0.427 |
| ETTm2 | 336 | 96 | 0.158 | 0.234 | 0.178 | 0.267 | 0.168 | 0.261 | 0.168 | 0.261 | 0.178 | 0.267 |
| ETTm2 | 336 | 192 | 0.214 | 0.267 | 0.234 | 0.308 | 0.226 | 0.300 | 0.227 | 0.299 | 0.234 | 0.304 |
| ETTm2 | 336 | 336 | 0.264 | 0.327 | 0.287 | 0.342 | 0.282 | 0.336 | 0.281 | 0.336 | 0.288 | 0.341 |
| ETTm2 | 336 | 720 | 0.360 | 0.383 | 0.370 | 0.395 | 0.361 | 0.386 | 0.361 | 0.387 | 0.366 | 0.387 |
| Exchange | 336 | 96 | 0.081 | 0.198 | 0.087 | 0.210 | 0.082 | 0.205 | 0.086 | 0.209 | 0.089 | 0.212 |
| Exchange | 336 | 192 | 0.176 | 0.297 | 0.181 | 0.390 | 0.175 | 0.302 | 0.180 | 0.308 | 0.189 | 0.314 |
| Exchange | 336 | 336 | 0.314 | 0.396 | 0.327 | 0.420 | 0.314 | 0.405 | 0.338 | 0.424 | 0.340 | 0.428 |
| Exchange | 336 | 720 | 0.712 | 0.606 | 0.916 | 0.719 | 0.783 | 0.632 | 0.918 | 0.717 | 0.886 | 0.704 |
| Weather | 336 | 96 | 0.140 | 0.177 | 0.159 | 0.209 | 0.154 | 0.204 | 0.159 | 0.210 | 0.161 | 0.207 |
| Weather | 336 | 192 | 0.187 | 0.227 | 0.202 | 0.245 | 0.199 | 0.246 | 0.202 | 0.249 | 0.201 | 0.247 |
| Weather | 336 | 336 | 0.234 | 0.266 | 0.252 | 0.290 | 0.250 | 0.287 | 0.252 | 0.288 | 0.242 | 0.283 |
| Weather | 336 | 720 | 0.302 | 0.311 | 0.322 | 0.336 | 0.326 | 0.393 | 0.325 | 0.392 | 0.321 | 0.337 |
| Mean | | | 0.307 | 0.345 | 0.332 | 0.373 | 0.321 | 0.364 | 0.329 | 0.368 | 0.338 | 0.373 |
| Difference | | | | | -8.21% | -8.25% | -4.66% | -5.46% | -7.01% | -6.73% | -10.01% | -8.22% |

Table 9: Extended hyperparameter search (number of Wavelet bands).

| Dataset | Context | H | 2 | | 5 | | 8 | |
|---|---|---|---|---|---|---|---|---|
| | | | MSE | MAE | MSE | MAE | MSE | MAE |
| ETTh1 | 336 | 96 | 0.357 | 0.383 | 0.355 | 0.374 | 0.354 | 0.380 |
| ETTh1 | 336 | 192 | 0.405 | 0.410 | 0.390 | 0.405 | 0.409 | 0.414 |
| ETTh1 | 336 | 336 | 0.393 | 0.409 | 0.386 | 0.412 | 0.390 | 0.411 |
| ETTh1 | 336 | 720 | 0.449 | 0.457 | 0.421 | 0.445 | 0.445 | 0.454 |
| ETTh2 | 336 | 96 | 0.271 | 0.367 | 0.262 | 0.321 | 0.265 | 0.322 |
| ETTh2 | 336 | 192 | 0.325 | 0.364 | 0.311 | 0.359 | 0.318 | 0.360 |
| ETTh2 | 336 | 336 | 0.312 | 0.363 | 0.315 | 0.364 | 0.314 | 0.369 |
| ETTh2 | 336 | 720 | 0.389 | 0.423 | 0.388 | 0.421 | 0.395 | 0.427 |
| ETTm1 | 336 | 96 | 0.288 | 0.330 | 0.288 | 0.321 | 0.287 | 0.329 |
| ETTm1 | 336 | 192 | 0.328 | 0.355 | 0.325 | 0.359 | 0.331 | 0.356 |
| ETTm1 | 336 | 336 | 0.358 | 0.378 | 0.356 | 0.377 | 0.360 | 0.377 |
| ETTm1 | 336 | 720 | 0.412 | 0.427 | 0.410 | 0.419 | 0.413 | 0.424 |
| ETTm2 | 336 | 96 | 0.165 | 0.249 | 0.158 | 0.234 | 0.162 | 0.245 |
| ETTm2 | 336 | 192 | 0.223 | 0.288 | 0.214 | 0.267 | 0.218 | 0.285 |
| ETTm2 | 336 | 336 | 0.272 | 0.321 | 0.264 | 0.327 | 0.273 | 0.319 |
| ETTm2 | 336 | 720 | 0.353 | 0.373 | 0.360 | 0.383 | 0.354 | 0.373 |
| *Best count (MSE)* | | | 1 | | **12** | | 3 | |
| *Best count (MAE)* | | | | 4 | | **11** | | 2 |

Table 10: Extended hyperparameter search (depth of tree decomposition).

| Dataset | Context | H | 1 | | 2 | | 3 | |
|---|---|---|---|---|---|---|---|---|
| | | | MSE | MAE | MSE | MAE | MSE | MAE |
| ETTh1 | 336 | 96 | 0.363 | 0.386 | 0.355 | 0.374 | 0.355 | 0.374 |
| ETTh1 | 336 | 192 | 0.416 | 0.416 | 0.390 | 0.405 | 0.407 | 0.411 |
| ETTh1 | 336 | 336 | 0.423 | 0.426 | 0.386 | 0.412 | 0.397 | 0.412 |
| ETTh1 | 336 | 720 | 0.495 | 0.482 | 0.421 | 0.445 | 0.446 | 0.461 |
| ETTh2 | 336 | 96 | 0.263 | 0.322 | 0.262 | 0.321 | 0.262 | 0.321 |
| ETTh2 | 336 | 192 | 0.321 | 0.361 | 0.311 | 0.359 | 0.316 | 0.359 |
| ETTh2 | 336 | 336 | 0.313 | 0.361 | 0.315 | 0.364 | 0.315 | 0.364 |
| ETTh2 | 336 | 720 | 0.387 | 0.417 | 0.390 | 0.421 | 0.386 | 0.419 |
| ETTm1 | 336 | 96 | 0.287 | 0.331 | 0.288 | 0.321 | 0.288 | 0.330 |
| ETTm1 | 336 | 192 | 0.327 | 0.355 | 0.325 | 0.359 | 0.330 | 0.355 |
| ETTm1 | 336 | 336 | 0.356 | 0.376 | 0.358 | 0.379 | 0.355 | 0.376 |
| ETTm1 | 336 | 720 | 0.412 | 0.427 | 0.410 | 0.419 | 0.413 | 0.423 |
| ETTm2 | 336 | 96 | 0.166 | 0.248 | 0.158 | 0.234 | 0.161 | 0.242 |
| ETTm2 | 336 | 192 | 0.221 | 0.286 | 0.214 | 0.267 | 0.219 | 0.284 |
| ETTm2 | 336 | 336 | 0.275 | 0.322 | 0.264 | 0.327 | 0.273 | 0.319 |
| ETTm2 | 336 | 720 | 0.355 | 0.374 | 0.352 | 0.372 | 0.353 | 0.373 |
| *Best count (MSE)* | | | 1 | | **13** | | 4 | |
| *Best count (MAE)* | | | | 5 | | **11** | | 6 |

Table 11: Extended hyperparameter search (overlap between temporal segments).

| Dataset | Context | H | 0 | | 8 | | 24 | |
|---|---|---|---|---|---|---|---|---|
| | | | MSE | MAE | MSE | MAE | MSE | MAE |
| ETTh1 | 336 | 96 | 0.359 | 0.383 | 0.355 | 0.374 | 0.358 | 0.382 |
| ETTh1 | 336 | 192 | 0.409 | 0.413 | 0.390 | 0.405 | 0.414 | 0.409 |
| ETTh1 | 336 | 336 | 0.416 | 0.405 | 0.386 | 0.412 | 0.397 | 0.411 |
| ETTh1 | 336 | 720 | 0.446 | 0.457 | 0.421 | 0.445 | 0.443 | 0.453 |
| ETTh2 | 336 | 96 | 0.265 | 0.322 | 0.262 | 0.321 | 0.271 | 0.327 |
| ETTh2 | 336 | 192 | 0.316 | 0.362 | 0.311 | 0.359 | 0.328 | 0.365 |
| ETTh2 | 336 | 336 | 0.319 | 0.367 | 0.315 | 0.364 | 0.318 | 0.365 |
| ETTh2 | 336 | 720 | 0.397 | 0.425 | 0.390 | 0.421 | 0.392 | 0.425 |
| ETTm1 | 336 | 96 | 0.288 | 0.332 | 0.288 | 0.321 | 0.287 | 0.331 |
| ETTm1 | 336 | 192 | 0.332 | 0.357 | 0.325 | 0.359 | 0.330 | 0.356 |
| ETTm1 | 336 | 336 | 0.360 | 0.377 | 0.358 | 0.379 | 0.360 | 0.378 |
| ETTm1 | 336 | 720 | 0.412 | 0.426 | 0.410 | 0.419 | 0.412 | 0.424 |
| ETTm2 | 336 | 96 | 0.162 | 0.245 | 0.158 | 0.234 | 0.163 | 0.246 |
| ETTm2 | 336 | 192 | 0.219 | 0.284 | 0.214 | 0.267 | 0.220 | 0.285 |
| ETTm2 | 336 | 336 | 0.271 | 0.319 | 0.264 | 0.327 | 0.272 | 0.319 |
| ETTm2 | 336 | 720 | 0.351 | 0.371 | 0.360 | 0.383 | 0.353 | 0.372 |
| *Best count (MSE)* | | | 0 | | **14** | | 2 | |
| *Best count (MAE)* | | | | 5 | | **11** | | 1 |

Table 12: Extended results (different context length).

| Dataset | Context | H | PRISM | | D-PAD | | Time Mixer | |
|---|---|---|---|---|---|---|---|---|
| | | | MSE | MAE | MSE | MAE | MSE | MAE |
| ETTh1 | 96 | 96 | 0.372 | 0.385 | 0.387 | 0.400 | 0.385 | 0.401 |
| ETTh1 | 96 | 192 | 0.432 | 0.415 | 0.453 | 0.436 | 0.440 | 0.428 |
| ETTh1 | 96 | 336 | 0.452 | 0.422 | 0.456 | 0.437 | 0.513 | 0.470 |
| ETTh1 | 96 | 720 | 0.463 | 0.447 | 0.469 | 0.463 | 0.501 | 0.480 |
| ETTh2 | 96 | 96 | 0.277 | 0.327 | 0.289 | 0.335 | 0.290 | 0.342 |
| ETTh2 | 96 | 192 | 0.347 | 0.373 | 0.359 | 0.381 | 0.382 | 0.401 |
| ETTh2 | 96 | 336 | 0.377 | 0.342 | 0.382 | 0.353 | 0.416 | 0.431 |
| ETTh2 | 96 | 720 | 0.401 | 0.423 | 0.408 | 0.429 | 0.419 | 0.440 |
| ETTm1 | 96 | 96 | 0.312 | 0.341 | 0.320 | 0.347 | 0.324 | 0.363 |
| ETTm1 | 96 | 192 | 0.368 | 0.370 | 0.375 | 0.375 | 0.369 | 0.389 |
| ETTm1 | 96 | 336 | 0.394 | 0.392 | 0.401 | 0.399 | 0.395 | 0.407 |
| ETTm1 | 96 | 720 | 0.459 | 0.437 | 0.479 | 0.438 | 0.459 | 0.446 |
| ETTm2 | 96 | 96 | 0.172 | 0.251 | 0.176 | 0.258 | 0.175 | 0.258 |
| ETTm2 | 96 | 192 | 0.237 | 0.294 | 0.243 | 0.301 | 0.239 | 0.306 |
| ETTm2 | 96 | 336 | 0.294 | 0.333 | 0.308 | 0.343 | 0.295 | 0.339 |
| ETTm2 | 96 | 720 | 0.392 | 0.395 | 0.397 | 0.402 | 0.393 | 0.396 |
| *Best count (MSE)* | | | **16** | | | | | |
| *Best count (MAE)* | | | | **16** | | | | |

