# OpenReview forum: "PRISM: A Hierarchical Multiscale Approach for Time Series Forecasting"
_ICLR.cc/2026/Conference — Submitted to ICLR 2026_

### Official Review · Reviewer_ri67 · 2025-10-23

**Soundness:** 3
**Presentation:** 3
**Contribution:** 2
**Rating:** 2
**Confidence:** 4

**Summary:**

The paper presents PRISM, a novel hierarchical multiscale model for time-series forecasting. PRISM constructs a binary tree over time while performing frequency decomposition (using wavelets or similar filters) at each node. It jointly learns temporal and frequency hierarchies and uses learnable importance scores to weight different frequency bands. By optimizing both forecasting and reconstruction losses, PRISM achieves strong interpretability, robustness, and state-of-the-art performance on multiple benchmarks.

**Strengths:**

1. The learnable importance scores provide insights into which frequency components drive predictions, adding transparency to the model’s behavior.
2. The paper is well-structured and easy to follow.

**Weaknesses:**

1. The binary partitioning strategy is manually defined rather than data-adaptive; this might limit flexibility for non-stationary or irregularly sampled signals.
2. **Dependence on pre-defined transforms**. The method relies on fixed wavelet or FFT bases. Learned or adaptive frequency decompositions could potentially capture more expressive features.
3. The benchmarks are still limited to standard ones. More various or big datasets should be put in place to demonstrate the effectiveness of the proposed method.
4. While the model provides interpretable frequency weights, the paper lacks formal evaluation or case studies illustrating interpretability in applications.

**Questions:**

1. How sensitive is the model to the choice of frequency basis (e.g., wavelet type)?
2. Could the tree depth be learned dynamically based on data complexity rather than fixed by design?
3. How does PRISM perform on highly irregular, non-periodic time series (e.g., event-driven or sparse data)?

---

> ### Author Response · Authors · 2025-11-20
> **Response to reviewer ri67 [Part 1]**
>
> Dear reviewer ri67,
>
>
> We sincerely thank you for your time and the positive comments. We appreciate your positive remarks regarding the paper’s clarity, well-structured presentation, sound methodology, strong interpretability and robustness. We are also grateful that you noted PRISM achieves strong performance (SOTA) across multiple benchmarks. To address the concerns you raised and clarify any potential misunderstandings, we provide detailed responses below:
>
>
> # W1 Binary partitioning vs. data-adaptive
> We would like to thank the Reviewer for this suggestion. Data-driven partitioning schemes have found widespread use in numerical methods, e.g. adaptive meshes in numerical solvers for differential equations. Transferring adaptive mesh methods to the domain of the time series forecasting is a promising future direction that we would be excited to explore in our future work.
>
> Currently, the data-adaptive methods in time-series forecasting are represented by different encoders. To compare our binary partitioning structure to a fully adaptive encoder, **we have performed an ablation in our original work where we have replaced it with an MLP encoder.**
>
> _Results Averaged Across 4 Horizons (H=96, 192, 336, 720 Samples), ETT Represents the Average of 4 ETT datasets._
> | Dataset | Ctx | PRISM MSE | MLP MSE | Improvement MSE | PRISM MAE | MLP MAE | Improvement MAE |
> |-|-|-|-|-|-|-|-|
> | ETT (4 datasets)    | 336 | **0.326** | 0.343 | 4.96% | **0.362** | 0.379 | 4.39% |
> | Exchange | 336 | **0.321** | 0.841 | 61.85% | **0.374** | 0.667 | 43.85% |
> | Weather | 336 | **0.216** | 0.237 | 8.87%.  | **0.245** | 0.278 | 11.70% |
>
>
> Across all six benchmarks, **PRISM consistently outperforms the MLP encoder in both MSE and MAE**. On the ETT datasets, PRISM gains 4.96% MSE improvement on ETT, with even larger gains on Exchange (61.85%) and Weather (8.87%), and similar trend with MAE improvements. These results show that PRISM’s binary partitioning provides a stronger inductive prior than a fully learnable encoder—preserving multiscale structure, preventing overfitting, and delivering reliably better predictions. The concerns regarding the flexibility and the ability to capture the non-stationary or irregularly sampled signals are further addressed in Q3 below with the GIFT evaluation datasets results.
>
>
>
>
> # W2 Dependence on predefined transforms
> As data-driven transformations have the potential of modeling the data better than fixed transformations through the use of specific properties of particular datasets, we have attempted, in our original submission, to **train a learnable filter or threshold** on the data as a first layer of our model (see below; also Table 3 and Supplementary Table 7 and 8).
>
> _Results Averaged Across 4 Horizons (H=96, 192, 336, 720 Samples), ETT Represents the Average of 4 ETT datasets._
> | Dataset  | Ctx | PRISM MSE | Learnable Weights MSE | Imp. MSE | Learnable Threshold MSE | Imp. MSE | PRISM MAE | Learnable Weights MAE | Imp. MAE | Learnable Threshold MAE | Imp. MAE |
> |-|-|-|-|-|-|-|-|-|-|-|-|
> | ETT (4 datasets)   | 336 | **0.326** | 0.343 | 5.08% | 0.355 | 8.14% | **0.362** | 0.381 | 5.02% | 0.389 | 7.03% |
> | Exchange | 336 | **0.321** | 0.399 | 19.61% | 0.376 | 14.69%  | **0.374** | 0.430 | 12.91% |0.415  | 9.71%|
> | Weather | 336 | **0.216** | 0.228 | 5.16% | 0.231 | 6.70%  | **0.245** | 0.264 | 7.01% | 0.269 | 8.66%|
>
> Across all three benchmarks, **the fixed Haar wavelet transform consistently outperforms both learnable weighting variants**. PRISM improves MSE by 11.08% on average over learnable weights and 10.36% over learnable thresholds, with similar MAE gains of 8.67% and 8.47%, respectively. This shows that the predefined Haar basis provides a more stable and effective multiscale representation, avoiding the collapse or overfitting issues seen in fully learnable decompositions.
>
> That matches the observations made previously in other fields: In audio, a non-learnable transformation (mel-spectrogram), was a long-standing best transformation until the data scaled to the point that has enabled learnable transformations to surpass it. In face recognition, even despite the data scale, it was found that the alignment of the faces based on non-learnable features from classical computer vision is necessary for the model to learn. Likewise, we find that for the current benchmarks, the non-learnable transformation performs better than a learnable one. However, should it be needed, our method is fully compatible with learnable transformations.

---

> ### Author Response · Authors · 2025-11-20
> **Response to reviewer ri67 [Part 2]**
>
> # W3 More variety of big datasets
> To address the Reviewer’s comment, we **have evaluated our model and its close competitors (D-PAD, DLinear) on the GIFT benchmark, containing 91 dataset**. Due to its size, this benchmark is ideal for providing extended evaluation. Going from 8 datasets of the original submission to 91 datasets in GIFT offers a more-than-tenfold expansion in the evaluation. On GIFT, among 91 datasets, we show that our method outperforms the competitors in the MSE on 61 datasets and in the MAE on 52 datasets (followed by the D-PAD that is the best on 21 and 25 datasets in MSE and MAE respectively). This result constitutes a 2-3 fold improvement over the D-PAD results.
>
> *Numbers Indicate How Many Datasets Each Model Wins on (Lower MSE/MAE).*
>
> | Model   | Overall MSE | Overall MAE | Irregular MSE | Irregular MAE | Aperiodic MSE | Aperiodic MAE | Incomplete MSE | Incomplete MAE | Anomaly MSE | Anomaly MAE | Nonstationary MSE | Nonstationary MAE | Drift MSE | Drift MAE |
> |-|-|-|-|-|-|-|-|-|-|-|-|-|-|-|
> | **PRISM (ours)** | **61** | **52** | **16** | **14** | **20** | **18** | **27** | **24** | **13** | **11** | **33** | **29** | **17** | **15** |
> | D-PAD   | 21 | 25 | 6 | 7 | 4 | 5 | 10 | 13 | 8 | 8 | 11 | 15 | 4 | 6 |
> | DLinear | 9 | 14 | 4 | 5 | 6 | 7 | 5 | 5 | 3 | 5 | 6 | 6 | 2 | 2 |
>
>
> # W4 Case study and formal evaluation of importance scores
> Following the Reviewer’s suggestion, **we added a case study and a formal evaluation of the importance scores in our model**. Section 4.3 (page 7) provides a case study for the importance scores on ETT datasets. The ETT datasets record the temperatures at 2 different power plants, each at 2 different timescales, thus enabling us to compare the importance scores (that we use for reweighting of the wavelet bands) on different yet related sources of data. We show that the importance scores are robust across tree segments, forecast lengths, numbers of wavelets, datasets recording similar events, and seeds. These results show that our model learns consistent data-driven representations despite the variability in the sources of similar data.
>
>
> **We also provide a formal evaluation of empirically derived importance scores**. Appendix A.2 (page 14) provides a formal analysis of the model’s optimal hyperparameters and their relation to the learned importance scores. We argue that the model learns to prioritize the frequencies from the highest available (determined by the dataset’s sampling rate) to the lowest admissible by known criteria from signal processing (determined by the context window length). We argue that the optimal depth of the time decomposition being equal to 2 likely tracks the trend of the data. In conclusion, we use the observations from this section to guide the hyperparameter selection for new datasets based on their loss function, sampling rate, and the context window length.
>
> # Q1 Sensitivity to frequency basis
> Following the Reviewer’s suggestion, **we have investigated the role of Haar’s wavelets in our model**. In the original submission, we have observed through an ablation study that, with the long context of H=336, Haar’s wavelets in our model outperform other frequency features on all tested datasets. To explain this observation, we have added the Supplementary Discussion on the role of Haar’s Wavelets in our model (Appendix A.1 of the updated manuscript), which we summarize below. **Briefly, Haar’s wavelets have an internal decomposition structure that is matched by the tree structure of our model**. That allows our model to seamlessly split and recombine the wavelet components. **In contrast, for other frequency bases (e.g. the FFT), there’s no such optimal matching with our model**. As a result, the temporal decomposition may introduce artifacts, e.g. the strong edge effects with the FFT. These edge effects disproportionately impact short time segments that, in the tree decomposition, represent high frequencies. Thus other filters underperform with our model.
>
> **There is, however, a regime where the wavelets are suboptimal. With short context lengths, the number of possible wavelets is limited as log_2 H, H is Horizons**. When the number of possible wavelets becomes small, it limits the forecasting abilities of the model. **In these cases (e.g. in some of the datasets in the GIFT-eval benchmark), we found that it’s optimal to use the exponential moving average instead.**
>
> Overall, we found that the wavelets are universally useful in long contexts, while, in shorter contexts, an exponential moving average is a better choice for the improvement of forecasting accuracy.

---

> ### Author Response · Authors · 2025-11-20
> **Response to reviewer ri67 [Part 3]**
>
> # Q2 Learnable tree depth
> To address the question of the optimal tree depth, we first **have performed the hyperparameter sweep over different tree depths** (table below)
>
> ### Sweep 1 — Depth of Tree (L)
> _Results Averaged Across 4 Horizons (H=96, 192, 336, 720 Samples) and the 4 ETT Datasets. Note Ctx=336._
> | Depth of the Tree | 1 | 2 | 3 | 4 |
> |-|-|-|-|-|
> | MSE | 0.336 | **0.326** | 0.330 | 0.331 |
> | MAE | 0.368| **0.362** | 0.364 | 0.365 |
> | Time (s) | 1.530 | 3.188 | 5.666 | 10.779  |
> | GPU Memory (MB) | 2444   | 3859 | 6939 | 12001 |
>
>
> We found that, for most of the considered settings, **the optimal number of tree layers was equal to 2**. Moreover, we found that, with the higher numbers of tree layers, the model’s run time and the required memory grows significantly, making the depth of 2 the optimal choice across those benchmark datasets. Splitting the time series into 2 components may be sufficient because it allows the model to identify the trend of the time series, while its periodic structure is captured by the wavelets.
>
> **The data-driven flexibility in the number of tree levels, however, can be implemented through the mechanisms that are already a part of our model**. Specifically, our model performs data-driven band selection through the importance scoring. Thus, one may add an excessive split to the tree (e.g. use 3 levels instead of 2) and, should it be unnecessary for the forecast, the model would decrease its importance scores, effectively reducing the number of tree splits.
>
> # W1, Q3 Irregular / non-periodic time series
> To investigate the robustness of our model under irregular/nonperiodic time series, we have, following the Reviewer’s suggestion, **performed additional evaluation on the GIFT benchmark, containing 91 dataset**. Due to its size, this benchmark is ideal for providing extended evaluation including evaluation on multivariate data. **Here we used the 91 dataset of the GIFT benchmark to analyze the performance of our model on irregular, aperiodic, incomplete (missing values),  anomalous, non-stationary, and slowly-drifting time series**. To this end, we selected datasets from the GIFT benchmark using their attached descriptions. We found that our model outperforms the D-PAD and DLinear, our closest competitors, in all 6 of these categories. That is, our method is the best on irregular data (best on 16 datasets vs. 6 for D-PAD), on aperiodic data (best on 20 datasets vs. 4 for D-PAD), on incomplete data (best on 27 datasets vs. 10 for D-PAD), on anomalous data (best on 13 datasets vs. 8 for D-PAD), on nonstationary data (best on 33 datasets vs. 11 for D-PAD), and on drifting data (best on 17 datasets vs. 4 for D-PAD). We present these results below.
>
> _Numbers Indicate How Many Datasets Each Model Wins on (Lower MSE/MAE)._
> | Model   | Overall MSE | Overall MAE | Irregular MSE | Irregular MAE | Aperiodic MSE | Aperiodic MAE | Incomplete MSE | Incomplete MAE | Anomaly MSE | Anomaly MAE | Nonstationary MSE | Nonstationary MAE | Drift MSE | Drift MAE |
> |---------|-------------|-------------|----------------|----------------|----------------|----------------|-----------------|-----------------|--------------|--------------|--------------------|--------------------|-----------|-----------|
> | **PRISM (ours)** | **61** | **52** | **16** | **14** | **20** | **18** | **27** | **24** | **13** | **11** | **33** | **29** | **17** | **15** |
> | D-PAD            | 21     | 25     | 6      | 7      | 4      | 5      | 10     | 13     | 8      | 8      | 11     | 15     | 4      | 6      |
> | DLinear          | 9      | 14     | 4      | 5      | 6      | 7      | 5      | 5      | 3      | 5      | 6      | 6      | 2      | 2      |

---

### Official Review · Reviewer_8cUd · 2025-10-30

**Soundness:** 2
**Presentation:** 2
**Contribution:** 2
**Rating:** 4
**Confidence:** 4

**Summary:**

The paper proposes PRISM, a time-series forecaster that builds a unified time–frequency hierarchy. Concretely: the model (i) performs binary time partitioning with overlap to form a tree, (ii) applies a time–frequency decomposition (default: Haar DWT) at each node, (iii) computes band importance weights via summary statistics → 2-layer MLP → softmax, and (iv) optimizes a joint loss that couples forecasting (future) with reconstruction (past). Experiments on 8 datasets × 4 horizons report SOTA or competitive results (best MSE in 17/32 settings; best MAE in 18/32), with extensive ablations showing the contribution of the tree encoder, wavelets, importance MLP, reconstruction loss, and residual connections. The motivation is that real-world series exhibit multi-scale behavior (global trends, local fluctuations, and intermediate scales), so hierarchical representations should align long-term structure and fine-scale variability.

**Strengths:**

* Clear problem framing and gap statement. The paper argues that prior work typically builds hierarchy in only time or only frequency, or mixes domains without a reconstructable shared hierarchy.
* Coherent architecture. Overlapped binary splits (time) + band partition (frequency) + learnable band routing + an explicit reconstruction path form a consistent design.
* Broad empirical coverage and ablations. Results across 32 settings, with component-wise ablations showing 5–14% average performance drops when removing key pieces (tree depth, wavelets, importance MLP, reconstruction loss, residual connections).
* Efficiency and interpretability. Training-time comparisons (e.g., ETTh1–96: 10 epochs in 65s) and band-importance visualizations support practical utility and model introspection.

**Weaknesses:**

* (Primary) Limited conceptual novelty relative to recent multiscale “decompose–mix” lines. The high-level philosophy—multiscale decomposition and mixing—strongly overlaps with recent TimeMixer-style approaches. The paper does cite such work in Related Work (e.g., Ref. [20]), but the manuscript does not clearly establish a qualitative leap beyond “engineered combination” of known ideas (time hierarchy + frequency filters + learned weighting + auxiliary reconstruction). Claimed distinction is a reconstructable, shared time–frequency tree, yet the empirical section lacks head-to-head, controlled comparisons designed to isolate scenarios where this specific design strictly dominates competing multiscale methods.
* Wavelet superiority appears under-analyzed. Table-2 shows average gains over FFT/EMA/DoG/MCD, but there is no conditioned analysis clarifying when wavelets lose/win based on spectral characteristics, periodicity, or multivariate correlations.
* Robustness in realistic settings is thin. The paper focuses on public benchmarks; it lacks systematic tests under missing values, anomalies, distribution shift/drift, or longer non-stationary horizons.
* Hyperparameter sensitivity is under-reported. No systematic sweep over overlap (o), tree depth, or number of bands (K) to reveal accuracy/time/memory trade-offs and boundary effects of the cross-fade concatenation.

**Questions:**

1. Differentiate from TimeMixer-style work with controlled, apples-to-apples tests. Under identical pipelines/tuning/resources, can you provide direct, multi-dataset head-to-head results and analyses showing where and why the reconstructable time–frequency tree and band-importance routing deliver significant, consistent gains? Please include long-context, non-stationary, and low-resource regimes.
2. When are wavelets the right choice? Offer a data-property ↔ filter mapping (e.g., by periodicity, noise spectrum, cross-channel dependencies). If possible, include learned basis experiments to test whether fixed Haar is limiting or optimal across conditions.
3. Hyperparameter sensitivity. Provide thorough sweeps for overlap (o), tree depth, and band count (K), reporting accuracy/time/memory and any bleeding/edge artifacts due to overlap and cross-fade.
4. Role of the reconstruction loss. Visualize the bias–variance trade-off between reconstruction and forecasting (e.g., gradients/importance by band as the reconstruction-loss weight varies). Do larger reconstruction weights ever suppress informative high-frequency bands?
5. Robustness. Add evaluations for missingness, anomalies, covariate shift, and long-term drift, comparing to strong linear/transformer/multiscale baselines.

---

> ### Author Response · Authors · 2025-11-20
> **Response to 8cUd [Part 1]**
>
> Dear reviewer 8cUd,
>
> We sincerely thank you for your time, interest in our work, and detailed actionables.
>
> # W1, Q1 Difference with TimeMixer
> While there are conceptual similarities between our model and TimeMixer, our paper focuses on the few critical differences that, we show, lead to our model’s improved performance across settings. **Unlike TimeMixer that performs deterministic downsampling in the time domain** and season-trends mixing, **our model performs the decomposition that’s hierarchical both in time and frequency**. This way, we apply the multiscale-decompose-and-mix idea in not one but two domains simultaneously which, to our knowledge, is novel.
>
> **For the direct comparison** of the two methods as proposed by the Reviewer, **we performed additional experiments with TimeMixer**. First, we evaluated TimeMixer in the long-context setting on the same tasks that we used to compare all other models. We found that, while TimeMixer has shown a strong performance, our method has performed even better in all tested settings (summary below; also Table 1 of the updated manuscript).
>
>
> _Results Averaged Across 4 Horizons (H=336, 192, 336, 720 Samples), ETT Represents the Average of 4 ETT datasets._
> | Dataset  | Ctx | PRISM MSE | TimeMixer MSE | Improvement MSE | PRISM MAE  | TimeMixer MAE | Improvement MAE |
> |-|-|-|-|-|-|-|-|
> | ETT (4 datasets)         | 336 | **0.326** | 0.361 | 9.62% | **0.362** | 0.391 | 7.49% |
> | Traffic     | 336 | **0.393** | 0.412 | 4.67%  | **0.252** | 0.288 | 12.41% |
> | Electricity | 336 | **0.155** | 0.164 | 5.34%  | **0.245** | 0.257 | 4.68%  |
> | Exchange    | 336 | **0.321** | 0.413 | 22.38% | **0.374** | 0.425 | 11.89% |
> | Weather     | 336 | **0.216** | 0.230 | 6.20%  | **0.245** | 0.271 | 9.33% |
>
> Next, **as TimeMixer was originally tested on shorter context windows, we have performed an additional experiment with forecasting on the context of 96 time points**. Here, our model has also outperformed TimeMixer in all the tested settings (summary below; also Supplementary Table 12 of the updated manuscript).
>
>
> _Results Averaged Across 4 Horizons (H=96, 192, 336, 720 Samples), ETT Represents the Average of 4 ETT datasets._
>
> | Dataset  | Ctx | PRISM MSE | TimeMixer MSE | Improvement MSE | PRISM MAE | TimeMixer MAE | Improvement MAE |
> |-|-|-|-|-|-|-|-|
> | ETT (4 datasets)   | 96  | **0.359** | 0.375 | 4.12% | **0.372** | 0.394 | 5.56% |
>
>
> As the main difference between TimeMixer and our model is in additional hierarchical decomposition of the data in the frequency domain, these results speak to the utility of joint hierarchical decomposition in both time and frequency.

---

> ### Author Response · Authors · 2025-11-20
> **Response to 8cUd [Part 2]**
>
> # W2, Q2 Wavelets vs. FFT etc
> Following the Reviewer’s suggestion, we have investigated the role of Haar’s wavelets in our model. In the original submission, we have observed through an ablation study that, with the long context of H=336, Haar’s wavelets in our model outperform other frequency features on all tested datasets. To explain this observation, we have added the Supplementary Discussion on the role of Haar’s Wavelets in our model (Appendix A.1 of the updated manuscript), which we summarize below. **Briefly, Haar’s wavelets have an internal decomposition structure that is matched by the tree structure of our model.** That allows our model to seamlessly split and recombine the wavelet components. In contrast, **for other frequency bases (e.g. the FFT), there’s no such optimal matching with our model.** As a result, the temporal decomposition may introduce artifacts, e.g. the strong edge effects with the FFT. These edge effects disproportionately impact short time segments that, in the tree decomposition, represent high frequencies. Thus other filters underperform with our model.
>
> **There is, however, a regime where the wavelets are suboptimal. With short context lengths, the number of possible wavelets is limited as log_2 H**, H is context length. When the number of possible wavelets becomes small, it limits the forecasting abilities of the model. **In these cases (e.g. in some of the datasets in the GIFT-eval benchmark), we found that it’s optimal to use the exponential moving average instead.**
>
> In our original submission, we performed learned basis experiments and trained a learnable filter or threshold on the data (see below; also Table 3 and Supplementary Table 7 and 8). Across six benchmarks, the fixed Haar wavelet transform consistently outperforms both learnable weights and filters.
>
> _Results Averaged Across 4 Horizons (H=96, 192, 336, 720 Samples), ETT Represents the Average of 4 ETT datasets._
> | Dataset  | Ctx | PRISM MSE | Learnable Weights MSE | Imp. MSE | Learnable Threshold MSE | Imp. MSE | PRISM MAE | Learnable Weights MAE | Imp. MAE | Learnable Threshold MAE | Imp. MAE |
> |-|-|-|-|-|-|-|-|-|-|-|-|
> | ETT (4 datasets)   | 336 | **0.326** | 0.343 | 5.08% | 0.355 | 8.14% | **0.362** | 0.381 | 5.02% | 0.389 | 7.03% |
> | Exchange | 336 | **0.321** | 0.399 | 19.61% | 0.376 | 14.69%  | **0.374** | 0.430 | 12.91% |0.415  | 9.71%|
> | Weather | 336 | **0.216** | 0.228 | 5.16% | 0.231 | 6.70%  | **0.245** | 0.264 | 7.01% | 0.269 | 8.66%|
>
> PRISM improves MSE by 11.08% on average over learnable weights and 10.36% over learnable thresholds, with similar MAE gains of 8.67% and 8.47%, respectively. This shows that the predefined Haar basis provides a more stable and effective multiscale representation, avoiding the collapse or overfitting issues seen in fully learnable decompositions. Overall, we found that the wavelets are universally useful in long contexts, while, in shorter contexts, an exponential moving average is a better choice for the improvement of forecasting accuracy.

---

> ### Author Response · Authors · 2025-11-20
> **Response to 8cUd [Part 3]**
>
> # W4, Q3 Hyperparameters
> To investigate the model’s stability w.r.t. hyperparameters, upon Reviewer’s request, **we added the hyperparameter sweeps** to the text (see the summary below; also Supplementary Tables 9, 10, and 11 of the updated manuscript). Additionally, in these tables **we provide the information regarding the time and memory use** in each of the analyzed settings.
>
> ### Sweep 1 — Depth of Tree
> _Results Averaged Across 4 Horizons (H=96, 192, 336, 720 Samples) and the 4 ETT Datasets. Note Ctx=336._
> | Depth of the Tree | 1 | 2 | 3 | 4 |
> |-|-|-|-|-|
> | MSE | 0.336 | **0.326** | 0.330 | 0.331 |
> | MAE | 0.368| **0.362** | 0.364 | 0.365 |
> | Time (s) | 1.530 | 3.188 | 5.666 | 10.779  |
> | GPU Memory (MB) | 2444   | 3859 | 6939 | 12001 |
>
> ### Sweep 2 – Overlap
> _Results Averaged Across 4 Horizons (H=96, 192, 336, 720 Samples) and the 4 ETT Datasets. Note Ctx=336._
> | Overlap | 0 | 8 | 24 | 168 |
> |-|-|-|-|-|
> | MSE | 0.333 | **0.326** | 0.332 | 0.334 |
> | MAE | 0.365 | **0.362** | 0.366 | 0.366 |
> | Time (s) | 3.959 | 4.389 | 4.661 | 5.181 |
> | GPU Memory (MB) | 4602 | 4187 | 4995 | 5338 |
>
> ### Sweep 3 – Number of Wavelet Components
>
> _Results Averaged Across 4 Horizons (H=96, 192, 336, 720 Samples) and the 4 ETT Datasets. Note Ctx=336._
>
> | Wavelet Components| 2 | 5 | 8 |
> |-|-|-|-|
> | MSE | 0.331 | **0.326** | 0.331 |
> | MAE | 0.369 | **0.362** | 0.365 |
> | Time (s) | 1.695 | 3.276 | 9.482  |
> | GPU Memory (MB) | 2409 | 3906 | 5307 |
>
> From these sweeps, we make three conclusions as follows.
> - First, the **models are stable w.r.t. the tree depth, number of wavelet bands, and the overlap** between the temporal segments: The MSE values saturate after the initial drop. Thus, setting deliberately large values for these hyperparameters is a viable strategy for the training on new datasets.
> - Second, the **parameters used for the experiments in our paper (5 wavelet bands, tree depth = 2, and the overlap of 8) are optimal**.
> - Third, the **time and memory consumption grows significantly after 5 wavelet bands and/or 2 tree layers**. Thus, **the parameters that are used in our work are optimal both for the model’s accuracy and speed**.
>
> To find why these parameters are optimal, we have added Appendix A.2. where **we provide a formal analysis of the model’s optimal hyperparameters and their relation to the learned importance scores**. We argue that the model learns to prioritize the frequencies from the highest available (determined by the dataset’s sampling rate) to the lowest admissible by known criteria from signal processing (determined by the context window length). We argue that the optimal depth of the time decomposition being equal to 2 likely tracks the trend of the data. In conclusion, **we use the results from this section to guide the hyperparameter selection for new datasets based on their loss function, sampling rate, and the context window length**.

---

> ### Author Response · Authors · 2025-11-20
> **Response to 8cUd [Part 4]**
>
> # Q4 Role of the reconstruction loss
> We found the reconstruction loss is helpful to stabilize the validation process and assists in the early stopping mechanism. Specifically, when we trained and validated our model on the combination of the forecasting and the reconstruction loss, the validation loss curve became smoother, making the early stopping less prone to noise. At the same time, as long as the reconstruction auxiliary loss is applied with any regularization coefficient, we have observed no bias-variance tradeoff between the reconstruction and the forecasting targets.
>
> # W3, Q5 Robustness under missing values etc
> To investigate the robustness of our model under missing values, we have, following the Reviewer’s suggestion, **performed additional evaluation on the GIFT benchmark, containing 91 dataset**. Due to its size, this benchmark is ideal for providing extended evaluation including evaluation on multivariate data. Here **we used the 91 dataset of the GIFT benchmark to analyze the performance of our model on irregular, aperiodic, incomplete (missing values),  anomalous, non-stationary, and slowly-drifting time series**. To this end, we selected datasets from the GIFT benchmark using their attached descriptions. We found that our model outperforms the D-PAD and DLinear, our closest competitors, in all 6 of these categories. That is, our method is the best on irregular data (best on 16 datasets vs. 6 for D-PAD), on aperiodic data (best on 20 datasets vs. 4 for D-PAD), on incomplete data (best on 27 datasets vs. 10 for D-PAD), on anomalous data (best on 13 datasets vs. 8 for D-PAD), on nonstationary data (best on 33 datasets vs. 11 for D-PAD), and on drifting data (best on 17 datasets vs. 4 for D-PAD). We present these results below.
>
> | Model   | Overall MSE | Overall MAE | Irregular MSE | Irregular MAE | Aperiodic MSE | Aperiodic MAE | Incomplete MSE | Incomplete MAE | Anomaly MSE | Anomaly MAE | Nonstationary MSE | Nonstationary MAE | Drift MSE | Drift MAE |
> |-|-|-|-|-|-|-|-|-|-|-|-|-|-|-|
> | **PRISM (ours)** | **61** | **52** | **16** | **14** | **20** | **18** | **27** | **24** | **13** | **11** | **33** | **29** | **17** | **15** |
> | D-PAD            | 21     | 25     | 6      | 7      | 4      | 5      | 10     | 13     | 8      | 8      | 11     | 15     | 4      | 6      |
> | DLinear          | 9      | 14     | 4      | 5      | 6      | 7      | 5      | 5      | 3      | 5      | 6      | 6      | 2      | 2      |

---

> > ### Comment · Reviewer_8cUd · 2025-11-24
> >
> > Thank you for the thorough response. The concerns I previously raised have now been fully clarified, and I appreciate the authors' efforts in strengthening the manuscript and additional experiments. I am satisfied with the revisions and will raise my score from 4 to 6.

---

> > > ### Author Response · Authors · 2025-11-24
> > >
> > > Thank you for the thoughtful feedback and the updated score! We’re pleased to hear that the revisions and additional results addressed your concerns. We truly appreciate your time and effort!

---

### Official Review · Reviewer_9JZA · 2025-11-03

**Soundness:** 4
**Presentation:** 4
**Contribution:** 3
**Rating:** 8
**Confidence:** 3

**Summary:**

This paper makes progress on unifying forecasting architectures using temporal hierarchies with methods using frequency modeling. The architecture splits the time series into temporal segments as a binary tree and applies a frequency filtering step at each hierarchy, along with residual connections at the frequency filtering steps. The final derived representations at the base of the binary tree are merged together using learned weights and a FFN is applied to generate the final predictions. The representations learn better due to the auxiliary reconstruction loss.

Experimental results on popular benchmark datasets show improved performance compared to most baselines except for the D-PAD baseline which performs closely to the proposed method.

**Strengths:**

- The paper presents an interesting technique to enhance hierarchical temporal architectures with frequency filtering. Frequency filtering seems to be an essential component of time series forecasting helping identify cyclical patterns in the dataset.
- Experimental results show that the method performs strongly compared to compared to most baselines, and performs closely to D-PAD which is a more complicated architecture.
- The paper is well presented and the ideas are quite clear. The experimental results are strengthened by the ablation study.

**Weaknesses:**

- The main weakness of the paper is that the method performs closely to D-PAD and isn't a significant improvement compared to D-PAD, however D-PAD is a much involved architecture compared to the proposed method.
- While, the paper evaluates on the most popular univariate datasets, some more complex datasets could be a newer addition such as M4 and wikipedia. While they are multi-variate datasets, they could help prevent overfitting of techniques on the existing datasets.

**Questions:**

- Why are the residual connections only in the frequency filters module? Are they not useful in the other layers?

---

> ### Author Response · Authors · 2025-11-20
> **Response to reviewer 9JZA**
>
> Dear reviewer 9JZA:
>
> Thank you for your  time and high valuation of our work, and actionable suggestions.
>
> # W1 Comparison with D-PAD
> While, aligned with the Reviewer’s comment, we value the simplicity and the interpretability of our model, it also outperforms the D-PAD stronger in different settings. **To address this comment, we have performed an additional experiment with the context length of 96. We show that our model outperforms the D-PAD both in MSE and MAE on all the ETT datasets.** We provide the full results in Supplementary Table 12 of the updated manuscript.
>
> _Results Averaged Across 4 Horizons (H=96, 192, 336, 720 Samples), ETT Represents the Average of 4 ETT Datasets._
>
> | Dataset | Ctx | PRISM MSE | D-PAD MSE | Improvement MSE | PRISM MAE | D-PAD MAE | Improvement MAE |
> |-|-|-|-|-|-|-|-|
> | ETT (4 datasets)  | 96 | **0.359** | 0.369 | 2.61% | **0.372** | 0.381 | 2.46% |
>
>
> # W2 Additional multivariate datasets
> Following the Reviewer’s suggestion, **we have performed an additional evaluation on the GIFT benchmark, containing 91 dataset. Due to its size, this benchmark is ideal for providing extended evaluation including evaluation on multivariate data** and the M4 dataset. Going from 8 datasets of the original submission to 91 datasets in GIFT offers a more-than-tenfold expansion in the evaluation. On GIFT, among 91 datasets, we show that our method outperforms the competitors in the MSE on 61 datasets and in the MAE on 52 datasets (followed by the D-PAD that is the best on 21 and 25 datasets in MSE and MAE respectively). This result constitutes a 2-3 fold improvement over the D-PAD results, addressing another concern previously raised by the Reviewer.
>
> _Numbers indicate how many datasets each model wins on (lower MSE/MAE)._
>
> | Best count  | PRISM | D-PAD | DLinear |
> |-|-|-|-|
> | **MSE**      | **61**| 21 | 9   |
> | **MAE**      | **52**| 25 | 14 |
>
> **Please find the extended table for this dataset in our general response*.*
>
> # Q1 Residual connections
> In our original model, we have a long-range residual connections bypassing the model. The residual connections are not useful in the other layers. For example, the forecasting module itself is an MLP, so a residual layer around it won’t apply. As for the frequency filter modules themselves, we have investigated, in the original text, different ways of placing residual connections and found that the ones applied to individual filter blocks lead to the best forecasting results. We provide the full results in Table 3 and Supplementary Table 6.

---

### Author Response · Authors · 2025-11-20
**Summary of Revisions**

We thank the Reviewers for their time and comments.

We are pleased that the Reviewers have found our approach interesting (9JZA, “interesting technique to enhance hierarchical temporal architectures”), coherent (8cUd, “coherent architecture”), and novel (ri67, “a novel hierarchical multiscale model”). We are grateful to see that the Reviewers have highlighted our model’s SOTA performance (9JZA, 8cUd,  ri67, e.g. “state-of-the-art performance on multiple benchmarks”), strong interpretability (8cUd, ri67, e.g. “learnable importance scores provide insights into which frequency components drive predictions”), and comprehensive ablations  (9JZA, 8cUd, e.g. “broad empirical coverage and ablations”) as strengths of our work. We are excited that all the Reviewers found our writing clear, well-presented, and easy to follow.

**Based on the Reviewer’s requests, we have made the following changes to the updated manuscript (highlighted in blue).**
## Evaluation on **91 new dataset**
**Section 4.2** (page 5) adds **evaluation on the GIFT benchmark, containing 91 dataset**, which is ideal for providing extended evaluation requested by Reviewers 9JZA and 8cUd **including evaluation on multivariate data** and the M4 dataset. The 91 datasets in GIFT offer a more-than-tenfold expansion of our previous submission. On GIFT, we show that our method outperforms the competitors in the MSE on 61 datasets and in the MAE on 52 datasets (followed by the D-PAD that is the best on 21 and 25 datasets in MSE and MAE respectively). This result constitutes a 2-3 fold improvement over the D-PAD results, addressing a concern previously raised by the Reviewer 9JZA.
| Dataset              | PRISM MSE | PRISM MAE | D-PAD MSE | D-PAD MAE | DLinear MSE | DLinear MAE |
|----------------------|-----------|-----------|-----------|-----------|-------------|-------------|
| bitbrains (7 datasets)           | **3.88E+06**  | 5.03E+02  | 4.17E+06  | 5.11E+02  | 4.37E+06    | **4.50E+02**    |
| bizitobs (11 datasets)            | **3.38E+08**  | 7.69E+03  | 3.47E+08  | **7.67E+03**  | 3.57E+08    | 7.75E+03    |
| car_parts/M/short    | **1.56E+00**  | 6.07E-01  | 1.59E+00  | 6.56E-01  | 1.61E+00    | **5.78E-01**    |
| covid_deaths/D/short | **1.32E+08**  | **2.68E+03**  | 2.05E+08  | 3.71E+03  | 3.29E+08    | 4.60E+03    |
| electricity (7 datasets)         | **2.41E+11**  | 6.30E+04  | 2.43E+11  | **5.45E+04**  | 2.41E+11    | 6.35E+04    |
| ett1 (8 datasets)                | **9.62E+05**  | **3.04E+02**  | 1.13E+06  | 3.26E+02  | 1.33E+06    | 3.37E+02    |
| ett2 (8 datasets)                 | **4.85E+07**  | **2.90E+03**  | 5.15E+07  | 2.98E+03  | 5.62E+07    | 3.10E+03    |
| hierarchical (2 datasets)         | 6.75E+02  | 1.28E+01  | **6.41E+02**  | **1.21E+01**  | 8.10E+02    | 1.37E+01    |
| hospital             | **1.54E+06**  | **4.55E+02**  | 1.54E+06  | 4.56E+02  | 1.54E+06    | 4.55E+02    |
| jena_weather (8 datasets)         | 2.15E+05  | 2.59E+02  | **2.05E+05**  | **2.52E+02**  | 2.18E+05    | 2.64E+02    |
| kdd_cup_2018 (4 datasets)         | **3.47E+03**  | **4.07E+01**  | 3.48E+03  | 4.11E+01  | 3.79E+03    | 4.10E+01    |
| loop_seattle (6 datasets)         | **1.34E+02**  | **6.45E+00**  | 4.88E+02  | 1.35E+01  | 1.48E+02    | 6.89E+00    |
| m4 (6 datasets)                  | **5.77E+08**  | **6.03E+03**  | 5.91E+08  | 6.03E+03  | 6.27E+08    | 6.14E+03    |
| m_dense (4 datasets)              | **3.43E+05**  | **3.66E+02**  | 3.74E+05  | 3.85E+02  | 4.11E+05    | 4.00E+02    |
| restaurant           | 5.18E+02  | 1.71E+01  | **5.03E+02**  | **1.68E+01**  | 1.01E+03    | 2.57E+01    |
| saugeen (3 datasets)             | **9.58E+02**  | **1.91E+01**  | 1.64E+03  | 2.57E+01  | 1.20E+03    | 2.17E+01    |
| solar (8 datasets)               | **1.85E+06**  | **3.83E+02**  | 4.21E+06  | 7.52E+02  | 2.03E+06    | 4.14E+02    |
| sz_taxi (3 datasets)             | 1.61E+02  | 9.58E+00  | 1.61E+02  | 9.57E+00  | **1.60E+02**    | **9.54E+00**    |
| temperature_rain     | 4.84E+02  | 1.31E+01  | **4.37E+02**  | **1.18E+01**  | 4.55E+02    | 1.25E+01    |
| us_births (3 datasets)            | **7.71E+06**  | **2.09E+03**  | 1.62E+10  | 5.39E+04  | 1.43E+07    | 2.82E+03    |
| *Best count (MSE)*   | 15        | --        | 4         | --        | 1          | --          |
| *Best count (MAE)*   | --        | 11        | --        | 6         | --         | 3          |

---

> ### Author Response · Authors · 2025-11-20
> **Summary of revisions [continued]**
>
> ## Evaluaion on anomalous / aperiodic data
> **Section 4.2** (page 5) uses GIFT to **analyze the performance of our model on irregular, aperiodic, incomplete (missing values), anomalous, non-stationary, and slowly-drifting time series**, as requested by Reviewers 8cUd and ri67. To this end, we selected datasets from the GIFT benchmark using their attached descriptions. We found that our model outperforms the D-PAD and DLinear, our closest competitors, in all 6 of these categories.
>
> *Numbers indicate how many datasets each model wins on (lower MSE/MAE).*
> | Model   | Overall MSE | Overall MAE | Irregular MSE | Irregular MAE | Aperiodic MSE | Aperiodic MAE | Incomplete MSE | Incomplete MAE | Anomaly MSE | Anomaly MAE | Nonstationary MSE | Nonstationary MAE | Drift MSE | Drift MAE |
> |-|-|-|-|-|-|-|-|-|-|-|-|-|-|-|
> | **PRISM (ours)** | **61** | **52** | **16** | **14** | **20** | **18** | **27** | **24** | **13** | **11** | **33** | **29** | **17** | **15** |
> | D-PAD   | 21 | 25 | 6 | 7 | 4 | 5 | 10 | 13 | 8 | 8 | 11 | 15 | 4 | 6 |
> | DLinear | 9 | 14 | 4 | 5 | 6 | 7 | 5 | 5 | 3 | 5 | 6 | 6 | 2 | 2 |
> ## Comparison with TimeMixer, the closest model
> **Section 4.1** (page 5) now **compares our model to TimeMixer, a similar model** that performs the multiscale decomposition in the time domain. Outperforming this model across benchmarks in a direct comparison supports the value of joint hierarchical decomposition in time and frequency.
> ## Case study: Stability of features
> **Section 4.3** (page 7) provides a **case study for the importance scores on ETT datasets**, as requested by Reviewer ri67. The ETT datasets record the temperatures at 2 power plants, each at 2 timescales, thus enabling us to compare the importance scores (that we use for reweighting of the wavelet bands) on different yet related sources of data. We show that the importance scores are robust across tree segments, forecast lengths, numbers of wavelets, datasets recording similar events, and seeds. These results show that our model learns consistent data-driven representations despite the variability in the sources of similar data.
> ## Comaprison of Wavelets vs FFT etc
> **Appendix A.1** (page 13) describes **how different feature bases (e.g., Wavelets vs. FFT) can be combined with our model**, as asked by Reviewers 8cUd and ri67. We show that the high performance of our method is supported by the unique structural pairing of our hierarchical splits in time and the Haar’s wavelets that use the same temporal decomposition scheme. We argue that other frequency bases are not as compatible with the hierarchical time decomposition: For example, the FFT would accumulate edge effects when computed on many short segments.
> ## Analysis of hyperparameters
> **Appendix A.2** (page 14) provides **analysis of the model’s optimal hyperparameters** requested by Reviewer ri67. We argue that the model prioritizes the frequencies from the highest available (determined by the dataset’s sampling rate) to the lowest admissible (determined by the context window length). These insights guide the hyperparameter selection for new datasets based on their loss function, sampling rate, and the context window length.
>
> **Tables 9, 10, 11** contain the **hyperparameter sweeps**, including the overlap, depth of trees and number of the components. We have now added these results to the text based on the request made by the Reviewer 8cUd.

---

### Meta-Review · Area_Chair_BQ1T · 2026-01-03

**Summary:**

The reviews are predominantly negative with the most expressing dissatisfaction about the weakness of the proposed method. Consequently,9JZA shows limited comparison;
8cUd expresses concerns about limited conceptual novelty, which is strongly overlaps with recent TimeMixer-style approaches. ri67 concerns about the effectivess of proposed method like manual strategy and reliance on fixed transforms. Although the
Although the authors have provided detailed reponses to the concerns, only part of concerns have been solved. Based on the limited contributions and strong overlap with previous TimeMixer, therefore, I tend to recommend the rejection of this paper in its current form.

**Reviewer Concerns:**

Most of concers of 8cUd and 8cUd  may have been solved. For others, part of concerns haven been solved.

**Reviewer Scores:**

8cUd may increase the score from 4 to 6.

---

### Decision · Program_Chairs · 2026-01-26

Reject